# Functional parcellation of mouse visual cortex using statistical techniques reveals response-dependent clustering of cortical processing areas

**Mari Ganesh Kumar**[1]☯*, **Ming Hu**[2]☯*, **Aadhirai Ramanujan**[1], **Mriganka Sur**[2]‡*, **Hema A. Murthy**[1]‡*

**1** Department of Computer Science and Engineering, Indian Institute of Technology Madras, Chennai, Tamil Nadu, India, **2** Department of Brain and Cognitive Sciences, Massachusetts Institute of Technology, Cambridge, Massachusetts, United States of America

☯ These authors contributed equally to this work.
‡These authors are joint senior authors on this work.
* mari@cse.iitm.ac.in (MGK); ming_hu@mit.edu (MH); msur@mit.edu (MS); hema@cse.iitm.ac.in (HAM)

**Data Availability Statement:** The paper presents results on two datasets, one collected using two-photo imaging and another collected using wide-

## Abstract

The visual cortex of the mouse brain can be divided into ten or more areas that each contain complete or partial retinotopic maps of the contralateral visual field. It is generally assumed that these areas represent discrete processing regions. In contrast to the conventional input-output characterizations of neuronal responses to standard visual stimuli, here we asked whether six of the core visual areas have responses that are functionally distinct from each other for a given visual stimulus set, by applying machine learning techniques to distinguish the areas based on their activity patterns. Visual areas defined by retinotopic mapping were examined using supervised classifiers applied to responses elicited by a range of stimuli. Using two distinct datasets obtained using wide-field and two-photon imaging, we show that the area labels predicted by the classifiers were highly consistent with the labels obtained using retinotopy. Furthermore, the classifiers were able to model the boundaries of visual areas using resting state cortical responses obtained without any overt stimulus, in both datasets. With the wide-field dataset, clustering neuronal responses using a constrained semi-supervised classifier showed graceful degradation of accuracy. The results suggest that responses from visual cortical areas can be classified effectively using data-driven models. These responses likely reflect unique circuits within each area that give rise to activity with stronger intra-areal than inter-areal correlations, and their responses to controlled visual stimuli across trials drive higher areal classification accuracy than resting state responses.

## Author summary

The visual cortex has a prominent role in the processing of visual information by the brain. Previous work has segmented the mouse visual cortex into different areas based on

field imaging. The two-photon dataset is a public dataset and can be accessed from http://observatory.brain-map.org/visualcoding. The wide-field dataset used in this study can be accessed from https://doi.org/10.6084/m9.figshare.13476522.v1. For both datasets, the software to reproduce the results are given in https://github.com/CCBR-IITMadras/visual-cortex-response-classification.

**Funding:** Supported by National Institutes of Health (NIH) grants EY007023 and EY028219 (MS), and the Center for Computational Brain Research (CCBR), IIT Madras, N.R. Narayanamurthy Chair Endowment. Narayanamurthy Chair endowment (MS). The funders had no role in study design, data collection, and analysis, decision to publish, or preparation of the manuscript.

**Competing interests:** The authors have declared that no competing interests exist.

the organization of retinotopic maps. Here, we collect responses of the visual cortex to various types of stimuli and ask if we could discover unique clusters from this dataset using machine learning methods. The retinotopy based area borders are used as ground truth to compare the performance of our clustering algorithms. We show our results on two datasets, one collected by the authors using wide-field imaging and another a publicly available dataset collected using two-photon imaging. The proposed supervised approach is able to predict the area labels accurately using neuronal responses to various visual stimuli. Following up on these results using visual stimuli, we hypothesized that each area of the mouse brain has unique responses that can be used to classify the area independently of stimuli. Experiments using resting state responses, without any overt stimulus, confirm this hypothesis. Such activity-based segmentation of the mouse visual cortex suggests that large-scale imaging combined with a machine learning algorithm may enable new insights into the functional organization of the visual cortex in mice and other species.

## Introduction

Visual cortex of higher mammals can be segmented into different functional visual areas. Each area has a distinct representation of the visual field and presumably a unique contribution to visual information processing. Historically, the functions of multiple cortical visual areas have been studied in non-human primates, which have well-defined areal parcellations based on visual field representations [1–4]. In the past few years, it has become clear that the mouse visual cortex can also be divided into different visual areas based on retinotopic organization. Indeed, mice have emerged as important models for studying the structure, function, and development of visual cortical circuits owing to their size, cost, and amenability to genetic perturbations [5].

Different methods have been used to define retinotopically organized visual cortical areas of the mouse brain. These methods include electrophysiological recording of receptive field [6], intrinsic signal imaging of visual field maps [7–9], voltage-sensitive dye imaging [10], and two-photon calcium imaging of receptive fields and maps [11, 12]. These techniques rely on retinotopic maps within representations of the contralateral visual field to derive visual areas. Precise visual area boundaries can be identified based on the sign of visual field representations [1, 9, 13–18], based on the principle that adjacent visual areas share a common representation of either the vertical or horizontal meridian and have essentially mirror-imaged maps across the common border. However, it is not clear if each of these retinotopically defined regions also has a unique functional role in processing visual information. Here, we use data-driven classifiers for studying the six most reliably identified visual areas in mice, namely, primary visual cortex (V1), lateromedial area (LM), anterolateral area (AL), rostrolateral area (RL), anteromedial area (AM) and posteromedial area (PM).

In contrast to the classical approach of studying these different visual areas using their neuronal tuning properties to different stimuli, this paper attempts to study whether the visual area responses can be differentiated from each other for a given stimulus-set utilizing data-driven approaches. We use datasets obtained using wide-field and two-photon imaging. Previous studies have used data-driven approaches such as convolutional neural networks (CNNs) [19] and localized semi-nonnegative matrix factorization (LocalNMF) [20] to derive insights from mouse visual cortex responses obtained using wide-field imaging. In this work, we use principal component analysis (PCA) and linear discriminant analysis (LDA) as a dimension

reduction technique to define a subspace which discriminates between neurons/pixels from different visual areas.

The wide-field dataset consists of responses of visual cortex to stimuli such as drifting gratings (varying orientation, spatial frequency, and temporal frequency), and natural movies collected using single-photon wide-field imaging. This imaging technique enables us to acquire data from a large field of view, albeit at a single pixel, multiple-neuron, spatial resolution due to light scattering. The retinotopic border of each area gives a segmentation of visual areas, which is used as ground truth for machine learning models. Utilizing this ground truth information, data-driven models are built using supervised and semi-supervised approaches to identify the visual area boundaries.

The population responses are first projected to a lower dimension space using techniques such as principal component analysis (PCA) and linear discriminant analysis (LDA). The data-driven models are trained using the reduced dimension representation and tested on unseen data. We find that supervised data-driven approaches are able to identify visual area borders with high accuracy. This supervised pipeline is then applied to a publicly available dataset provided by the Allen Institute for Brain Science [21] (available from: http://observatory.brain-map.org/visualcoding). This dataset is collected using two-photon microscopy and consists of individual neuronal responses recorded from the six visual areas. With this dataset, we get an accuracy significantly better than random chance. Resting state or spontaneous responses, obtained without overt visual stimuli, from both datasets are also able to classify areal borders, though classification is better with trial-averaged visually driven responses. The findings suggest that the activity patterns of different visual areas can be used to reliably and accurately classify their borders. Correlation analyses indicate that intra-area correlations are significantly higher than inter-area correlations for both datasets, providing a basis for the classification. We further validate these results by removing retinotopic information gradually from the wide-field training data. The results indicate that different visual areas can be distinguished by statistical characteristics expressed in their visually-driven or resting state activity.

# 1 Materials and methods

## 1.1 Ethics statement

The experiments for collecting the wide-field dataset (Section 1.2.1) were carried out under protocols approved by MIT's Animal Care and Use Committee (Protocol Approval Number: 1020-099-23) and conform to NIH guidelines.

## 1.2 Datasets

This section briefly describes two datasets collected using wide-field and two-photon imaging, respectively. In both datasets, the mouse visual cortex was first partitioned into different visual areas using a retinotopic map [9]. The wide-field dataset was collected by the authors on awake, head-fixed mice, which transgenically express GCaMP6f or GCaMP6s. Since this dataset was collected using single-photon wide-field imaging, the spatial resolution is limited to the pixels of the microscopic image. The two-photon dataset is a public-dataset released by the Allen Institute for Brain Science. Unlike the wide-field imaging dataset, this dataset has individual neuronal responses recorded from different areas and mice.

**1.2.1 Wide-field dataset. Animals and surgery.** The dataset for this work was collected from five adult mice (>8 weeks old) of either sex. These mice expressed GCaMP6f or GCaMP6s in excitatory neurons of the forebrain. The mouse lines were generated by crossing Ai93 (for GCaMP6f) and Ai94 (for GCaMP6s) with Emx1-IRES-Cre lines from Jackson Labs.

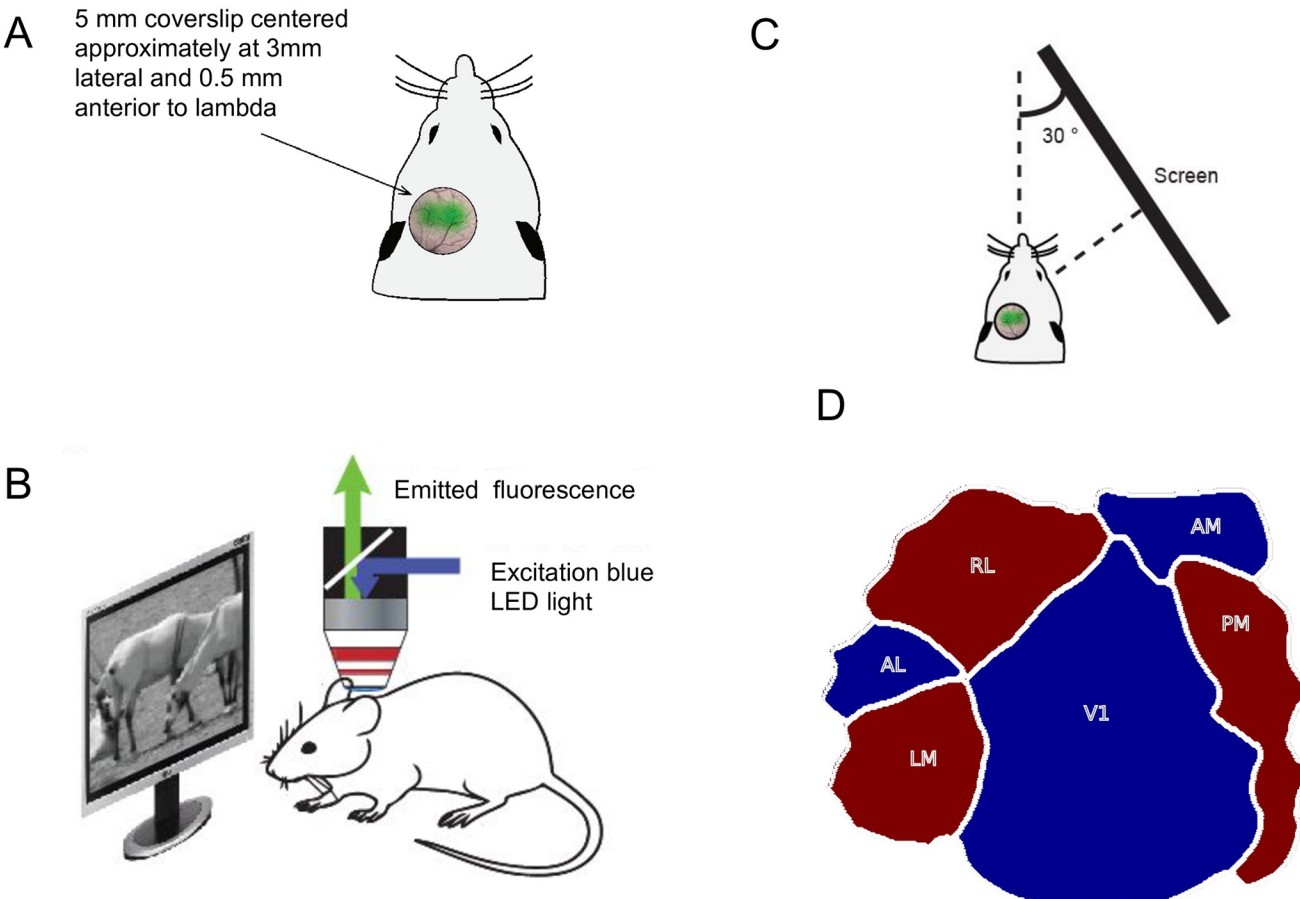

**Fig 1. Experimental setup. A**) Diagrammatic representation of a mouse prepared for wide-field imaging. **B**) Diagrammatic representation of the custom-made one-photon wide-field imaging setup along with the display screen. **C**) Display configuration relative to imaging the left cortex. **D**) Visual cortex map showing the sign of visual field representations along with areal boundaries for six visual areas used in this study.

The surgical procedure for implanting headplates and imaging window was similar to that previously described [22]. Mice were anesthetized using isoflurane (3% induction, 1.5-2% during surgery). A custom-built metal head-post was attached to the skull using dental cement, a 5 mm diameter craniotomy was performed over visual cortex of the left hemisphere, and a glass coverslip was glued over the opening (Fig 1A). Care was taken not to rupture the dura mater. The core body temperature was maintained at 37.5° **C** using a heating blanket (Harvard Apparatus). After recovery from surgery, mice were acclimatized to head fixation and then imaged while awake.

**Imaging and visual stimulation.** The imaging device used to prepare this dataset was a custom-made one-photon wide-field microscope. The light emitted from a blue LED (Thorlabs) was used to excite the GCaMP, and a monochrome CCD camera (Thorlabs) collected the emitted fluorescent signal (Fig 1B). Cortical responses were recorded at 1392 x 1040 resolution, with a spatial resolution of 200 pixels per mm of the cortex at a maximal frame rate of 20 Hz.

Visual stimuli were presented to head-fixed mice using a large display screen placed perpendicular to the right retina at an angle of 30 relative to the body axis of the animal. The visual display subtended 131° horizontal x 108° vertical at 12 cm eye-to-screen distance. The display was gamma-corrected, and placement was ensured to cover nearly the entire contralateral visual field (Fig 1C). The mean luminance of the screen was kept at $\approx 55 cd/m2$.

**Table 1. Summary of different stimuli shown to mice.**

| S. No | Stimuli Name | Description |
|---|---|---|
| 1 | Directions/ Orientation | 16 different sinusoidal gratings with varying directions from 0° to 360° with a step of 22.5°. The spatial and temporal frequencies were fixed at $0.03_{cycles/degree}$ and 3 Hz, respectively. Michelson contrast of 0.8 was used. |
| 2 | Spatial-Frequency | 5 different sinusoidal gratings with spatial frequency increasing exponentially from $0.01_{cycles/degree}$ to $0.16_{cycles/degree}$. The temporal frequency was fixed at 3 Hz. For each spatial frequency, the direction was varied from 0° to 360° with a step of 45°. Michelson contrast of 0.8 was used. |
| 3 | Temporal-Frequency | 5 different sinusoidal gratings with temporal frequency increasing exponentially from 0.5 Hz to 8 Hz. The spatial frequency is fixed at $0.03_{cycles/degree}$. For each temporal frequency, the direction was varied from 0° to 360° with a step of 45°. Michelson contrast of 0.8 was used. |
| 4 | Natural Movies | 4 different movies with natural scenes. For each movie, additional noisy versions were created by perturbing their spatial correlations, as demonstrated in [22]. |

Retinotopic mapping of the visual cortex was first performed using periodic moving bars with checker-board texture that were 14° wide and spanned the width or height of the monitor. The boundaries of the 6 core visual areas were defined according to procedures described in [9] (Fig 1D). In S1 Fig, we show the horizontal and vertical retinotopy for all five mice along with the area borders. These retinotopically defined boundaries were considered as ground truth for delineating visual areas based on responses to different visual stimuli.

Table 1 gives a summary of different stimuli that were presented. The stimulus set included drifting sinusoidal gratings of different orientations and movement directions, with varying spatial and temporal frequencies; and four natural movies chosen from the Van Hateren movie database [23]. For each movie, additional noisy versions were created by perturbing their spatial correlations, as demonstrated in [22].

The different stimuli mentioned in Table 1 were presented 10 times in a block design. In each block, a random permutation of different stimuli was used (for example, in case of directional stimuli, gratings drifting in different directions were presented in random order for each block or trial). The duration of each natural movie stimulus was 4 secs, and that of all the other stimuli was 2 secs. In between two consecutive stimuli, a grey and blank screen was shown for 2 secs to capture the baseline responses. All imaging was done with awake head-fixed mice at rest. In addition to the stimuli mentioned in Table 1, resting state responses were also collected for 15 mins while awake head-fixed mice rested in complete darkness.

**1.2.2 Two-photon dataset.** This dataset is a subset of the Allen Brain Observatory dataset, available publicly at http://observatory.brain-map.org/visualcoding), as a part of the Allen Mouse Brain Atlas [21]. This dataset has individual neuronal responses recorded from mice of different transgenic Cre-lines. This work uses the Cre-lines "Emx1-IRES" (the whole cortex), and "Nr5a1" (layer 4 specific), which has recordings from all the six visual areas.

The neuronal responses to three different natural movies from this dataset were used in our data-driven analysis. The duration of these three movies were 30, 30, and 120 secs, in order. It is to be noted that the set of natural movies presented in this dataset is different from the ones used for collecting the wide-field dataset. In addition to natural movie responses, we also use the spontaneous/resting state activity of the cells. Resting state activity in this dataset was collected for 5 mins with a plain grey screen.

For each transgenic line, the dataset had neuronal responses of various mice collected using four different session types. In each session, different stimuli were used. For Natural Movies 1, 3, and spontaneous activity, all the neurons that were recorded during "Session A" from a

 

**Table 2. Number of neurons available for analysis for each Cre-line and session from the Allen Institute dataset.**

| Cre-line | AL | LM | RL | AM | PM | V1 |
|---|---|---|---|---|---|---|
| Emx1-IRES (Session A) | 1235 | 1446 | 1963 | 241 | 536 | 2199 |
| Emx1-IRES (Session C2) | 1148 | 1238 | 2085 | 226 | 552 | 964 |
| Nr5a1 (Session A) | 178 | 256 | 1074 | 110 | 203 | 441 |
| Nr5a1 (Session C2) | 106 | 267 | 1023 | 115 | 234 | 149 |

particular transgenic line were selected for analysis. Similarly, for Natural Movie 2, all the neurons from "Session C2" were selected for analysis. Detailed information on different sessions, stimuli, and the data-collection procedure is given in [24]. Since this dataset consists of individual neuronal responses, the numbers of neurons from each area varied from session to session. In Table 2, we detail the number of neurons available from each area for the Cre-lines "Emx1-IRES" and "Nr5a1". Similar details for other Cre-lines in the Allen Brain Observatory dataset are shown in S1 Table and the corresponding results are given in S2 Table.

### 1.3 Feature extraction from neuronal responses

In a traditional setting, the selectivity of neurons to a given visual stimulus set is characterized by their tuning curves. The tuning curves are the averaged firing rate expressed as a function of one or more parameters describing stimuli. The problem with this approach is that the response of the neuron to a particular stimulus is defined just by a scalar value. This section proposes an alternate way to represent the selectivities of neurons using dimensionality reduction techniques such as principal component analysis (PCA) and linear discriminant analysis (LDA).

**1.3.1 Principal component analysis.** PCA is a statistical dimensionality reduction technique which can be used to find directions of maximum variability. PCA was first introduced in [25] as an analog of the principal axis theorem; it was later developed and proposed in [26]. PCA is used extensively in diverse fields from neuroscience to physics because it is a simple, non-parametric method of obtaining relevant information from complex datasets. Mathematically, PCA can be defined as an orthogonal linear transform from a set of possibly correlated bases into a set of orthogonal bases called principal components. The principal components are the eigenvectors of the covariance matrix obtained by solving the generalized eigenvalue problem.

$$\Sigma = E\left[(\mathbf{X} - \bar{\boldsymbol{\mu}})(\mathbf{X} - \bar{\boldsymbol{\mu}})^t\right] \tag{1}$$

$$\Sigma \times \mathbf{V} = \mathbf{V} \times \mathbf{D} \tag{2}$$

where $\mathbf{X}$ is the data matrix with each column corresponding to an instance of the data, $\bar{\boldsymbol{\mu}}$ is the mean of the data matrix, $\Sigma$ is the covariance matrix, $\mathbf{V}$ is the eigenmatrix containing eigenvectors of $\Sigma$, and $\mathbf{D}$ is a diagonal matrix of eigenvalues. Any given data instance $\mathbf{x}_i$ can be represented as a weighted sum of the principal components.

$$\mathbf{x}_i = \sum_{n=1}^{N} \alpha_{in} \mathbf{v}_k \tag{3}$$

where $N$ is the total number of principal components, the contribution of each principal component $\mathbf{v}_k$ to the $i^{th}$ example is given by $\alpha_{in}$. The principal component corresponding to the largest eigenvalue captures the maximum variance present in the dataset. For any given $K$, the

 

principal components corresponding to top $K$ eigenvalues weighted by the corresponding $\alpha_i$s can be shown to give minimum reconstruction error [27].

PCA is a general technique for reducing dimensionality. In the literature, it has been widely used for modeling neuronal responses [28–32]. In this paper, we use PCA to reduce the response length (time series) of a neuronal population to a lower dimension by accumulating statistics across neurons. The PCA features computed in this way can be used to represent the neuronal response in a lower dimension space.

**1.3.2 Linear discriminant analysis.**   LDA is a supervised dimension reduction technique. PCA finds a set of directions explaining the variance in the dataset without using any class labels. On the other hand, LDA uses the class labels to find a linear projection that discriminates them [33].

Let $C$ be the total number of visual areas. The scatter between different visual areas can be explained by the matrix $\Sigma_b$ as given in Eq 4

$$\Sigma_b = \frac{1}{C}\sum_{c=1}^{C}(\bar{\mu}_c - \bar{\mu})(\bar{\mu}_c - \bar{\mu})^t \tag{4}$$

where $\bar{\mu}_c$ is the mean of the responses of visual area $c$, and $\bar{\mu}$ is the mean of the entire dataset. The LDA projection matrix $\hat{W}$ is computed using scatter matrix $\Sigma_b$ and covariance matrix $\Sigma$ as given in Eq 5.

$$\hat{W} = \underset{W}{\mathrm{argmax}}\left|\frac{W^t \Sigma_b W}{W^t \Sigma W}\right| \tag{5}$$

Any given data instance $\mathbf{x}_i$ can be represented as a weighted sum of LDA components.

$$\mathbf{x}_i = \sum_{n=1}^{N}\beta_{in}\mathbf{w}_k \tag{6}$$

where $N$ is the total number of LDA dimensions chosen, the contribution of each LDA basis $\mathbf{w}_k$ to the $i^{th}$ example is given by $\beta_{in}$. $\mathbf{w}_k$ is the k-th LDA direction. Similar to PCA, LDA is also used to reduce the dimension of population neuronal responses. However, PCA is primarily a technique for data compression, whereas LDA projects to a sub-space where the discrimination between the classes is maximal. We specifically employed LDA using the visual area segmentation (obtained by retinotopy) to obtain the projection matrix.

## 2 Results

### 2.1 Supervised classification of mouse visual areas

In this experiment, the responses of visual areas to various stimuli were tested for signatures that discriminate among them using supervised models. To ensure that the results were not biased towards a classifier, different supervised classifiers, namely, parametric unimodal and multimodal Bayesian classifiers [27], neural networks [34] and SVM classifiers [35] were used to classify visual areas based on their responses.

The wide-field/two-photon calcium response for any given visual stimulus was first averaged across trials and converted to a lower dimension space using PCA followed by LDA (Section 1.3). It is to be noted that the stimulus information (Section 1.2) was only used to average across trials, and no other explicit information about the stimulus configuration or retinotopy was given as input (we cannot exclude the possibility that stimuli such as natural movies contain implicit retinotopic information). The reduced dimension feature vectors were then used to train different supervised classifiers.

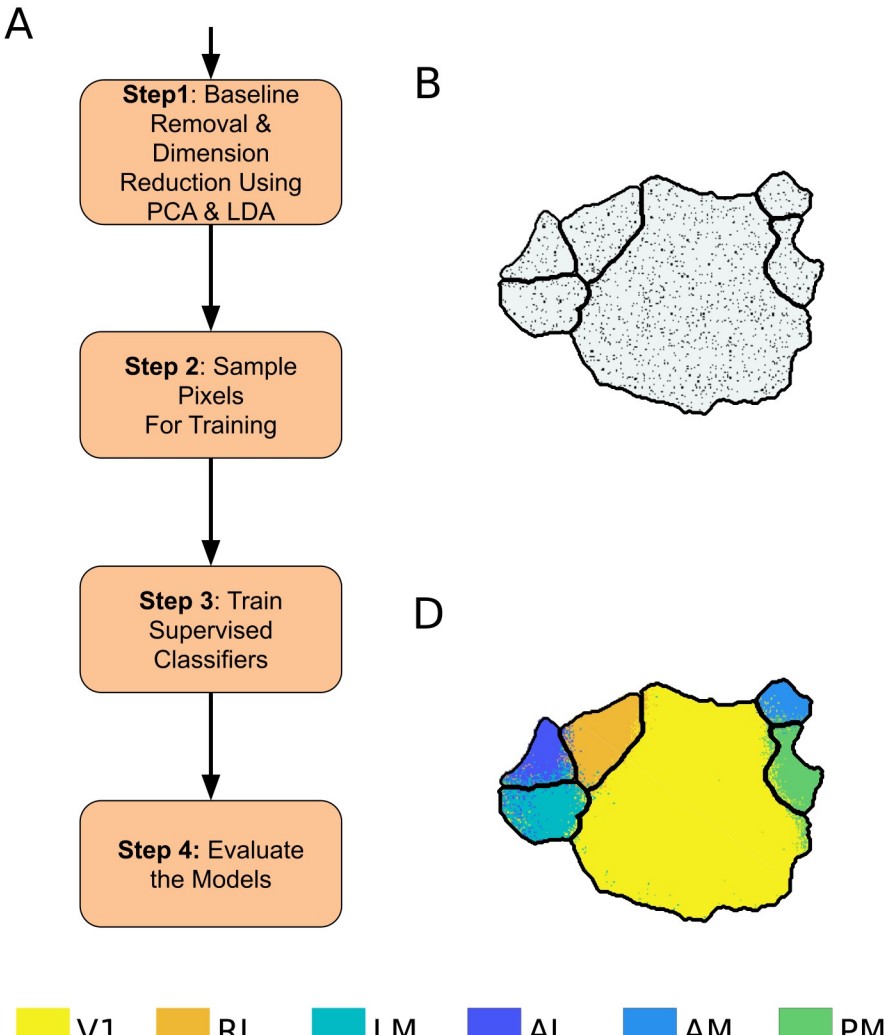

**Fig 2. Pipeline for supervised classification of visual areas. A**) Block diagram for supervised classification of visual cortex. **B**) The pixels chosen for training the classifiers are shown as black dots. **C**) Result of classifying visual cortex using the supervised GMM classifier. Boundaries in black denote the ground truth.

For the wide-field dataset, pixels in each visual area were divided into 'training' and 'test' sets. The models were trained using only about 5% of data from each area that were chosen randomly. This is because a high correlation was observed among the nearby pixels in wide-field data. With 5% sampling, these pixels are scattered such that their correlations are minimal. If the features of pixels within each area are similar, then it should be possible to classify them using the trained models.

The two-photon dataset captures responses of only few individual neurons from each area (Table 2). Hence, 5% of the total number of neurons from each area were not sufficient for training the supervised models. Thus, for the two-photon dataset, we randomly sampled 50% of the neurons from each area and kept the remainder for test. A general pipeline for the supervised model is given in Fig 2. The different classifiers used are described briefly below.

**2.1.1 Uni-modal Bayesian classifier.** In this method, visual areas are modeled using a parametric unimodal distribution. The equation for the Gaussian distribution is given below:

$$p(\mathbf{x}|\boldsymbol{\lambda}) = \mathcal{N}(\mathbf{x}|\bar{\boldsymbol{\mu}}, \boldsymbol{\Sigma})$$
$$= (2\pi)^{-d/2}|\boldsymbol{\Sigma}_k|^{-1/2} \times \tag{7}$$

$$exp\left\{-\frac{1}{2}(\mathbf{x}-\bar{\boldsymbol{\mu}})^T\boldsymbol{\Sigma}^{-1}(\mathbf{x}-\bar{\boldsymbol{\mu}})\right\} \tag{8}$$

where $\mathbf{x}$ is a D-dimensional random vector that describes the response of a pixel or neuron. $\boldsymbol{\lambda} = \{\bar{\boldsymbol{\mu}}$ and $\boldsymbol{\Sigma}\}$ are the parameters that describe the Gaussian distribution. For each visual area, the parameters are determined using maximum likelihood estimation (MLE). With the generative models trained using MLE, Bayes' rule is used for classification, where it is assumed that the priors are all equal.

$$\hat{A} = \operatorname*{argmax}_{1 \le k \le A} p(\mathbf{x}|\boldsymbol{\lambda}_k) \tag{9}$$

where $A$ is the total number of visual areas, $\lambda_k$ describes the model for the $k^{th}$ visual area, and $\hat{A}$ is the predicted visual area. Bayes classifier in this setup reduces to a maximum likelihood classifier (Eq 9). As the covariance matrix of each class is different, the boundaries between different areas can be a hyper-quadratic surface.

**2.1.2 Multimodal Bayesian classifier.** Gaussian mixture models (GMMs) are used to model visual areas in this method. GMMs are generative models. The motivation for using GMM is that it uses a multimodal probability density function to represent each visual area. GMMs can model any non-linear boundary between the visual areas in principle. A GMM is a density function which is a weighted sum of $M$ independent component densities given by the equation:

$$p(\mathbf{x}|\boldsymbol{\lambda}) = \sum_{k=1}^{M} w_k \mathcal{N}(\mathbf{x}|\bar{\boldsymbol{\mu}}_k, \boldsymbol{\Sigma}_k) \tag{10}$$

where $\mathbf{x}$ is a D-dimensional random vector that describes the neuronal response, $\mathcal{N}(\mathbf{x}|\bar{\mu}_k, \Sigma_k)$, $k = 1, \ldots, M$, are the unimodal component densities with mean $\bar{\boldsymbol{\mu}}_k$ and covariance $\Sigma_k$. $w_k$, denote the respective component weight. $\boldsymbol{\lambda} = \{w_k, \bar{\boldsymbol{\mu}}_k, \boldsymbol{\Sigma}_k\}_{k=1}^{M}$ defines the GMM for a particular visual area. The problem of fitting a GMM is an incomplete data problem. Hence, the mixtures need to be estimated iteratively using Expectation-Maximization (E-M). The procedure for training these parameters using E-M is detailed in [27]. A separate GMM is trained for each visual area by randomly sampling pixels from each visual area. For classifying the test data, the Bayes classifier given in Eq 9 was used. The number of mixtures for the GMMs is a hyperparameter and was estimated empirically.

**2.1.3 Support vector machine.** Let $D = \{\mathbf{x}_i, y_i\}_{i=1}^{N}$, where $\mathbf{x}_i \in R^d$ represent the training data, $y_i \in \{-1, 1\}$ represent the corresponding labels, $N$ is the total number of data points, and $d$ is the dimension of the feature vectors. In this setup, the support vector machine (SVM) finds a maximum separating hyperplane as follows:

$$f(\mathbf{x}) = \mathbf{w}^T\psi(\mathbf{x}) + b \tag{11}$$

where $\mathbf{w}$ is a normal vector to the separating hyperplane, and $b$ is the bias of the same. $\psi(.)$ is a transformation function from input feature space to kernel space. The optimization problem

for obtaining the separating hyperplane is given by:

$$\text{minimize} \quad \frac{1}{2}||\mathbf{w}||^2 + C\sum_{i=1}^{N}\xi_i \tag{12}$$

$$\text{subject to} \quad y_i[\mathbf{w}^T \psi(\mathbf{x}) + b] \geq 1 - \xi_i \quad \forall i = 1 \ldots N \tag{13}$$

$$\xi_i \geq 0 \quad \forall i = 1 \ldots N \tag{14}$$

where $C$ is a hyperparameter and $\xi_i$ is a slack variable that accounts for non-separable data problems. The details of the optimization algorithm can be found in [35]. A single SVM can solve only a binary class problem, a *one-against-one* approach was used to model the multi-class problem, and the final class label was determined using voting strategy. An open-source library LIBSVM [36] was used to implement visual area classifier using SVM.

**2.1.4 Artificial neural networks.** Artificial neural networks (ANN) are non-linear classifiers that can be trained to predict the area label from the response of the pixel. The advantage of using a feed-forward neural network over a conventional GMM is that the ANN is trained in a discriminative way. Let $\mathbf{o}_i = f(\mathbf{x}_i)$ be a non-linear function modeled by ANN, where $\mathbf{x}_i$ is the input vector representing the neuronal response and $f(.)$ is a differentiable non-linear function that can be modeled using ANN. Since this is a multi-class classification problem, softmax non-linear function is used as the activation function for the output layer. Softmax function outputs the probability of different classes. Therefore, the natural choice for the cost function is the cross-entropy between the target class labels and the output of the softmax function. The ANN is trained using backpropagation of the gradient of this cost function. A simple single hidden-layer ANN with 30 nodes was chosen to classify the neuronal response.

For each classifier, the rank-1 classification accuracy of the test data was used as the evaluation metric. The results were averaged across five random initializations of training data. The average and standard deviation of the classification accuracy obtained for the wide-field dataset are given in Table 3.

From the results of supervised classifiers in Table 3, it can be observed that the classifiers have similar performance across visual stimuli. Further, the unimodal Bayesian classifier gave the poorest classification accuracy, while the non-linear classifiers GMM and ANN gave the best performance.

**Table 3. Accuracy of supervised classification on wide-field data.** The results are averaged across random initializations. The entries denote "*% accuracy (± standard deviation)*".

| Mouse Number | Stimulus | Supervised | | | |
|---|---|---|---|---|---|
| | | **GMM** | **SVM** | **ANN** | **Bayes** |
| 1 | Directions | 94.3 (±0.63) | 94.1 (±0.16) | 94.3 (±0.32) | 93.0 (±0.48) |
| | Spatial-Frequency | 94.4 (±0.43) | 94.2 (±0.51) | 94.4 (±0.65) | 94.0 (±0.26) |
| | Temporal-Frequency | 90.8 (±0.92) | 90.2 (±0.86) | 90.8 (±0.81) | 90.1 (±0.73) |
| | Natural Movies | 97.0 (±0.13) | 96.2 (±0.27) | 97.6 (±0.44) | 92.7 (±0.50) |
| 2 | Natural Movies | 97.4 (±0.18) | 97.4 (±0.15) | 97.4 (±0.36) | 96.8 (±0.11) |
| 3 | Natural Movies | 97.2 (±0.14) | 97.0 (±0.28) | 97.1 (±0.08) | 96.4 (±0.12) |
| 4 | Natural Movies | 94.9 (±0.33) | 82.5 (±0.77) | 95.9 (±0.82) | 87.2 (±0.29) |
| | Resting State | 98.4 (±0.15) | 98.3 (±0.15) | 98.7 (±0.20) | 97.7 (±0.10) |
| 5 | Natural Movies | 96.1 (±0.38) | 83.9 (±0.52) | 97.4 (±1.01) | 84.5 (±0.50) |
| | Resting State | 96.7 (±0.32) | 96.9 (±0.14) | 97.8 (±0.17) | 95.4 (±0.35) |

We obtained areal boundaries by applying supervised GMM classification to neuronal population responses of different visual stimuli (Fig 3A). The visual areas predicted by the supervised models are color-coded. The area borders obtained by the classification are close to the retinotopic boundaries for all visual stimuli and all mice used. These results were verified to be consistent by training and testing responses of different mice to natural movie stimuli (Fig 3B).

Wide-field imaging captures the aggregated responses of hundreds of neurons, and pixels that are close together are highly correlated. Since the training data for the supervised model are sampled randomly from each visual area, the classification accuracy observed in Fig 3A and 3B can be an artifact of correlated responses. Consequently, this pipeline for the supervised classifiers was further verified with the two-photon dataset described in Section 1.2.2, which is obtained from individual neuron responses. The results are summarized in Table 4. Here, we obtain an average and a maximum accuracy of $\approx$ 58% and 70%, respectively.

In Fig 4, we show examples of confusion matrices, obtained using mouse M1 from wide-field dataset and Emx1-IRES Cre-line from two-photon dataset, respectively. In S2 Fig, we show the same for the entire dataset. For the wide-field dataset, responses from other areas were mostly predicted as V1 (Fig 4), which is not unexpected since V1 projects to each of the other areas. Following V1, the areas AL, LM and AM, PM had the next highest confusion. This is consistent with previous studies suggesting that these areas may constitute different processing streams [12, 37–39]. The confusion observed in the two-photon dataset were variable. Importantly, however, for both datasets, the majority of neurons were predicted correctly.

To demonstrate the significance of results obtained in Tables 3 and 4, we compared the results with two random classifiers. First, a random, unbiased six-faced die was considered. This random classifier will give a chance level accuracy of 16.67%, irrespective of the dataset. Secondly, we considered a six-faced die biased by the proportion of different area sizes (or the number of pixels/neurons used during training). For the wide-field dataset, this random classifier will give an average chance level accuracy of 37.6% (averaged across all five mice) and a maximum of 51.1% (for M1). Similarly, for dataset 2, this classifier will give an average chance accuracy of 26.7% and a maximum of 33.9% (for Nr5a1 Session C2). For both datasets, the results obtained in Tables 3 and 4 were much higher than the random classifiers. These results suggest that the responses are indeed discriminative between different areas.

The major difference between the two-photon and the wide-field dataset is that in the latter, the neighboring pixels can have correlated responses. To further test the effect of correlated responses in the wide-field dataset, the selection of training pixels was restricted to the center of visual areas. The training samples were limited to the circle formed with a sample radius from the center of each visual area. (first row of Fig 3D). The corresponding classification accuracies are reported in the second row of Fig 3D. Note that, despite the limited sampling, the classifiers were still able to classify the visual area accurately. These results thus indicate that "the responses of different visual areas to a range of visual stimuli can be used to reliably and accurately classify their borders".

These visual responses represent the net effect of feed-forward, local, long-range, and feedback circuits that drive neurons. We hypothesized that even the background or resting state responses, obtained without any overt visual stimulation, should contain the signatures required to identify each area. We tested this hypothesis by using resting state recordings from both datasets. In all experiments done with visual stimuli, the responses were averaged across trials to obtain stimulus-specific responses. However, in the resting state dataset, spontaneous responses changed with time without any explicitly defined trial structure. For the wide-field dataset, the non-averaged 800 secs of resting state responses excluding initial and final 50 secs of 15 mins recording sessions (to exclude non-stationary transients) were used in this analysis.

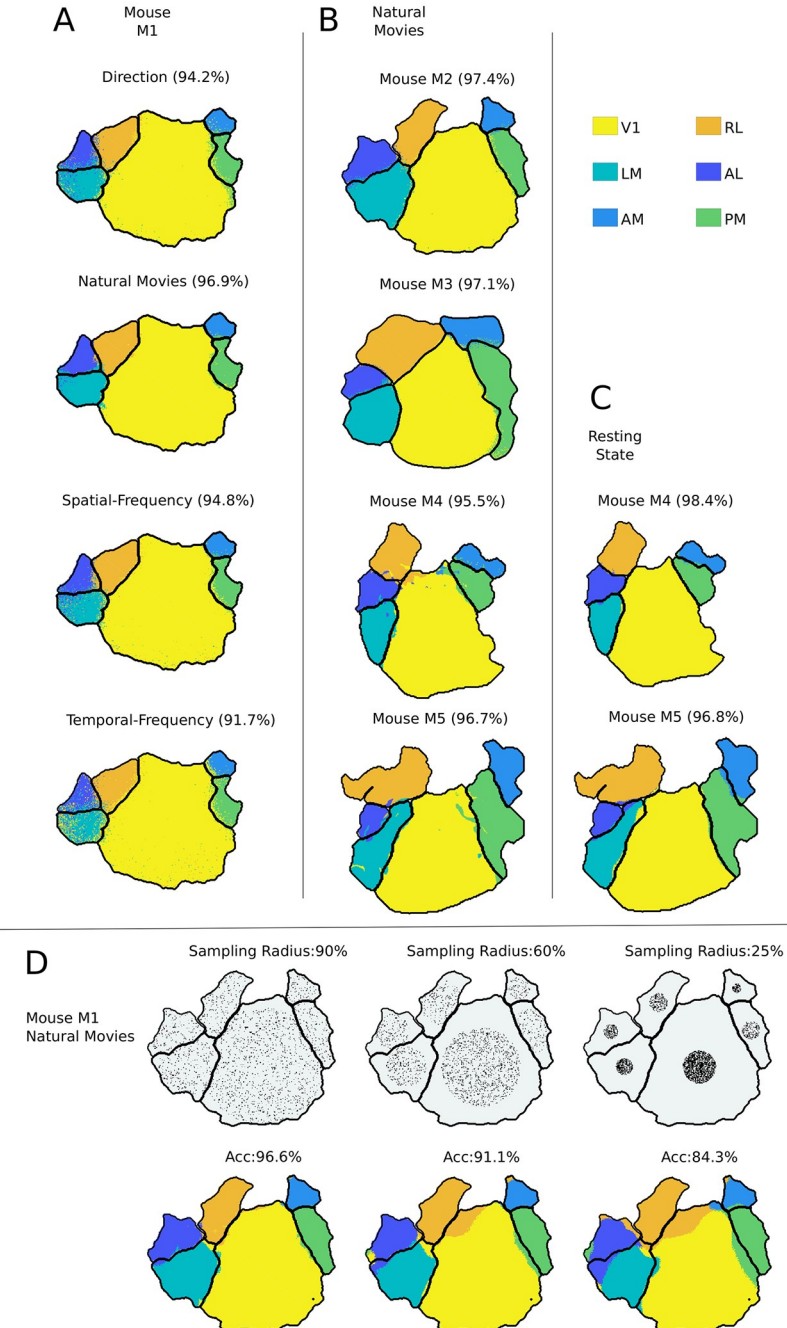

**Fig 3. Analysis of visual cortex responses to different stimuli using supervised GMM classifiers. A**, **B**) The boundaries obtained by classifying all the pixels using the GMM classifier. Each color represents the visual area identified by the classifier and the black boundary within the cortex corresponds to ground truth retinotopic boundaries. The values within the bracket denote the classification accuracy. In **A**, the results are compared across different visual stimuli. The title of the plot indicates the visual stimuli shown to the mice. In **B**, the supervised classifier is verified to be consistent across different mice for natural movie stimuli. **C**) Results on resting state responses for two mice. **D**) Pixels selected for training the supervised model are limited to center *x*% of the radius of the visual area. This *x*% is mentioned as sample radius in the title of the plots in the first row of **D**, and the pixel used for training the supervised model is shown as black dots. The corresponding classification boundaries are shown in the second row of **D**, and the "ACC" values denote the accuracy.

**Table 4. Accuracy of supervised classification on two-photon dataset.** The results are averaged across random initializations. The entries denote "*% accuracy (± standard deviation)*".

| Cre-line (Session) | Stimuli | Accuracy of Supervised Classifier | | | |
|---|---|---|---|---|---|
| | | GMM | SVM | ANN | Bayes |
| Emx1-IRES (Session A) | Natural Movie 1 | 54.6 (±0.75) | 56.8 (±0.65) | 56.6 (±0.88) | 54.3 (±0.70) |
| | Natural Movie 3 | 70.1 (±0.86) | 70.8 (±1.15) | 70.2 (±1.39) | 68.9 (±0.93) |
| | Resting State | 57.2 (±1.33) | 61.3 (±1.30) | 60.6 (±1.73) | 48.1 (±1.31) |
| Emx1-IRES (Session C2) | Natural Movie 2 | 52.6 (±0.58) | 55.2 (±0.70) | 55.0 (±0.65) | 52.0 (±0.18) |
| Nr5a1 (Session A) | Natural Movie 1 | 53.0 (±1.14) | 52.7 (±0.99) | 51.18 (±0.92) | 45.3 (±0.75) |
| | Natural Movie 3 | 59.6 (±1.19) | 61.0 (±0.98) | 60.2 (±0.89) | 54.9 (±1.58) |
| | Resting State | 45.6 (±2.89) | 53.8 (±1.18) | 52.7 (±1.10) | 44.3 (±0.93) |
| Nr5a1 (Session C2) | Natural Movie 2 | 55.1 (±1.20) | 56.2 (±1.76) | 54.3 (±1.45) | 46.3 (±0.87) |

The results of this analysis are shown in Fig 3C and show consistent high accuracy comparable to that obtained with visual stimuli (Table 3). Similarly, for the two-photon dataset, non-averaged 240 secs of resting state responses were used after excluding the initial and final 30 secs of total 5 mins recording from "Session A" of the dataset. Table 4 shows the result of classifying the resting state neuronal responses from dataset 2.

Irrespective of stimulus-driven or resting state responses, on both datasets, the proposed classification pipeline gave results that were much better than chance. The results of supervised classifiers suggest that the activity of each area has a specific statistical characteristic which can be used identify the area. As a control experiment, keeping the test data labels intact, we shuffled the training data labels randomly. For both the datasets, this led to accuracy of ≈ 14% to 18% (similar to the random chance level of 16% for 6 classes). This experiment shows that the observed results are not an artifact of the supervised approach. To further evaluate the influence of supervised labels in the results observed, in Section 2.2 we use a semi-supervised clustering approach for the wide-field dataset.

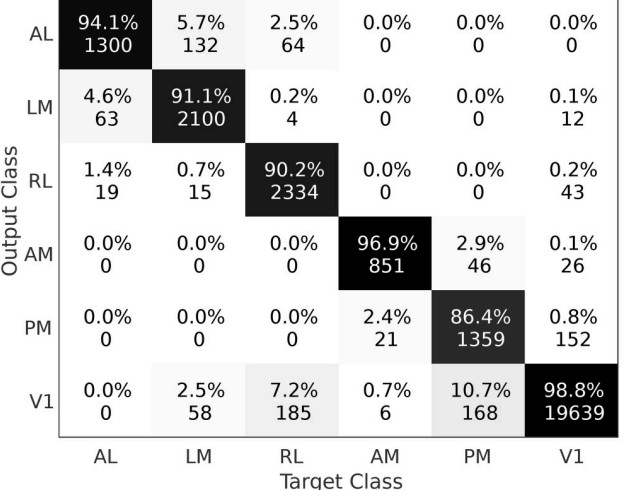

**Fig 4. Confusion matrices for test data obtained using supervised classifier.** The diagonal values denote the precision (in %) of each class. Off-diagonal values denotes the false prediction rate (in %) for the predicted class given the actual class. **A**) Confusion matrix obtained using responses of Mouse M1 and Natural Movie stimuli. **B**) Confusion matrix obtained using the Cre-line Emx1-IRES and Natural Movie 3 stimuli from dataset 2. In S2 Fig, we show the confusion matrices for all the remaining data.

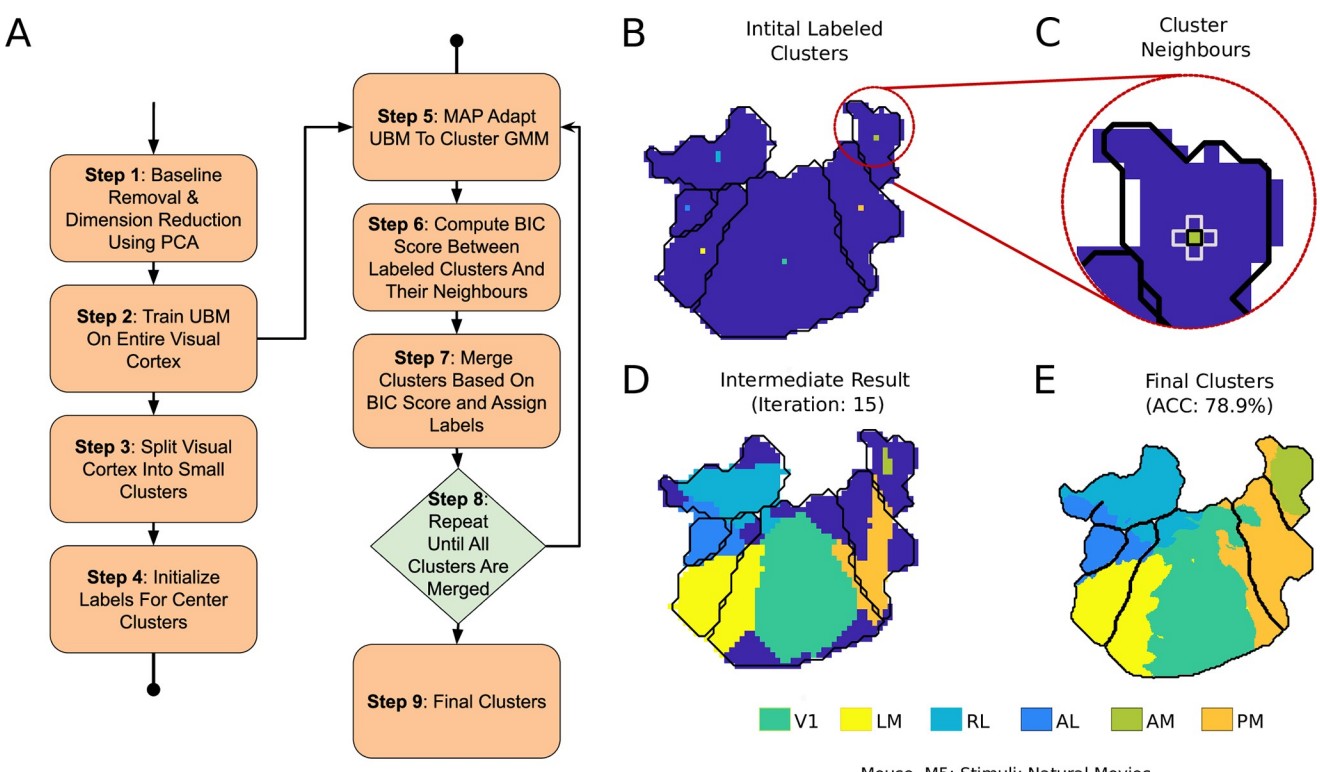

**Fig 5. Pipeline for semi-supervised clustering of visual areas. A**) Block diagram of the clustering steps. **B**) Initial clusters that are labeled using the retinotopic map. **C**) Neighbors of a labeled cluster for area *AM*. BIC score is computed between these neighbors, and closest few are merged every iteration. **D**) An intermediate step in the clustering process. **E**) Final clustering result with accuracy (ACC).

## 2.2 Semi-supervised clustering of mouse visual areas

In Section 2.1, the boundaries defined by retinotopy were used as ground truth, and the visual areas were tested for discriminative responses. Even after restricting the training data points to small regions at the center, reliable classification accuracy was obtained. In this Section, the supervised retinotopic information was further decreased to few pixels in the center of each visual area (shown in Fig 5B). In the supervised approach, 5% of pixels were randomly chosen from each visual area for training the classifier. In such a sampling, the number of pixels chosen for training is proportional to the size of the area. In semi-supervised clustering, the same amount of training data is used from each area.

In this approach of semi-supervised clustering, a special GMM called universal background model (UBM) was used. UBM is a model widely used for biometric verification, especially in speech [40]. UBM models the person-independent characteristics which can be used as a reference against the person-specific models. For example, in the case of speaker verification biometric, UBM is a speaker-independent GMM that is trained using speech samples obtained from a large set of speakers. Using maximum-a-posteriori (MAP) adaptation, the UBM can be adapted to be speaker-specific [41]. For our application, the UBM refers to a GMM trained using the responses of the entire visual cortex. Since the UBM is expected to model the entire visual cortex, a large number of Gaussians are required as compared to a GMM for individual visual areas.

The visual cortex was initially divided into small clusters of equal size such that they each had adequate data for MAP adaptation. Using MAP, UBM means were adapted to obtain cluster-specific GMMs. These cluster-specific GMMs are probability density functions that represent the responses of each cluster with UBM as the reference. The entire visual cortex was then hierarchically clustered, starting from the center clusters of each visual area (Fig 5B).

Let $a$ and $b$ represent the two neighboring clusters. The score for merging the clusters is calculated as follows:

$$S_{a,b} = log\ p(D|\lambda) - (\ log\ p(D_a|\lambda_a)\ +\ log\ p(D_b|\lambda_b)\ ) \tag{15}$$

where,

- $D_a$ and $D_b$ represent the data in each of the clusters. $D$ is the data of the combined cluster, i.e., $D_a \cup D_b$.

- $\lambda_a$ represents the parameters of the GMM obtained by MAP-adapting the UBM to $D_a$. Similarly, $\lambda_b$ and $\lambda$ are the parameters obtained by adapting the UBM to $D_b$ and $D$, respectively.

Eq 15 is the measure of the increase in likelihood after merging the models. It is a modified version of the Bayesian Information Criterion (BIC) [42] first proposed for clustering in [43]. For every iteration, a threshold of the score is determined, and the clusters are merged. Adaptive thresholding for every iteration eliminates the need for the penalty term given in [42]. The center-most cluster is the only supervised information given to the algorithm. The details of the semi-supervised clustering approach used in this paper are given below:

Step 1: The dimensions of wide-field responses are reduced using PCA (as described in Section 1.3).

Step 2: An UBM is trained using the response of the entire visual cortex.

Step 3: The entire visual cortex is divided into small grids of equal size.

Step 4: The center cluster/grid of each visual area is labeled using the retinotopic map. This step has been demonstrated in Fig 5B.

Step 5: The UBM is adapted to the labeled clusters and their neighbors to form cluster-specific-GMMs.

Step 6: The BIC score is calculated between the labeled clusters and their neighbors (as shown in Fig 5C).

Step 7: The BIC scores computed in Step 5 are sorted, and the top $x$% are merged. This $x$% starts with 80% and reduces to 20% as the clusters are labeled.

Step 8: Steps 4 to 6 are repeated until all the grids are labeled.

Step 9: Finally, to smoothen the boundaries, a supervised classifier is trained by sampling from the final clusters.

The result of clustering the visual areas using the semi-supervised approach is given in Table 5. The UBM modeled using the entire visual cortex response can be slightly different for different initialization, and it could affect the final clustering result. Hence, the clustering was performed using three different random initializations, and the mean and standard deviation of the obtained accuracy is given in Table 5. The boundaries obtained for different mice and stimuli are shown in Fig 6.

**Table 5. Accuracy of semi-supervised segmentation on wide-field dataset.** The results are averaged across random initializations of UBM. The entries denote "*% accuracy (± standard deviation)*".

| Mouse Number | Stimuli | Accuracy |
|---|---|---|
| 1 | Direction | 64.54 (±4.19) |
| | Spatial-Frequency | 56.3 (±3.42) |
| | Temporal-Frequency | 59.5 (±4.74) |
| | Natural Movies | 61.1 (±2.01) |
| 2 | Natural Movies | 71.1 (±0.87) |
| 3 | Natural Movies | 78.0 (±0.90) |
| 4 | Natural Movies | 78.0 (±3.43) |
| | Resting State | 72.1 (±1.71) |
| 5 | Natural Movies | 77.1 (±1.12) |
| | Resting State | 78.2 (±0.40) |

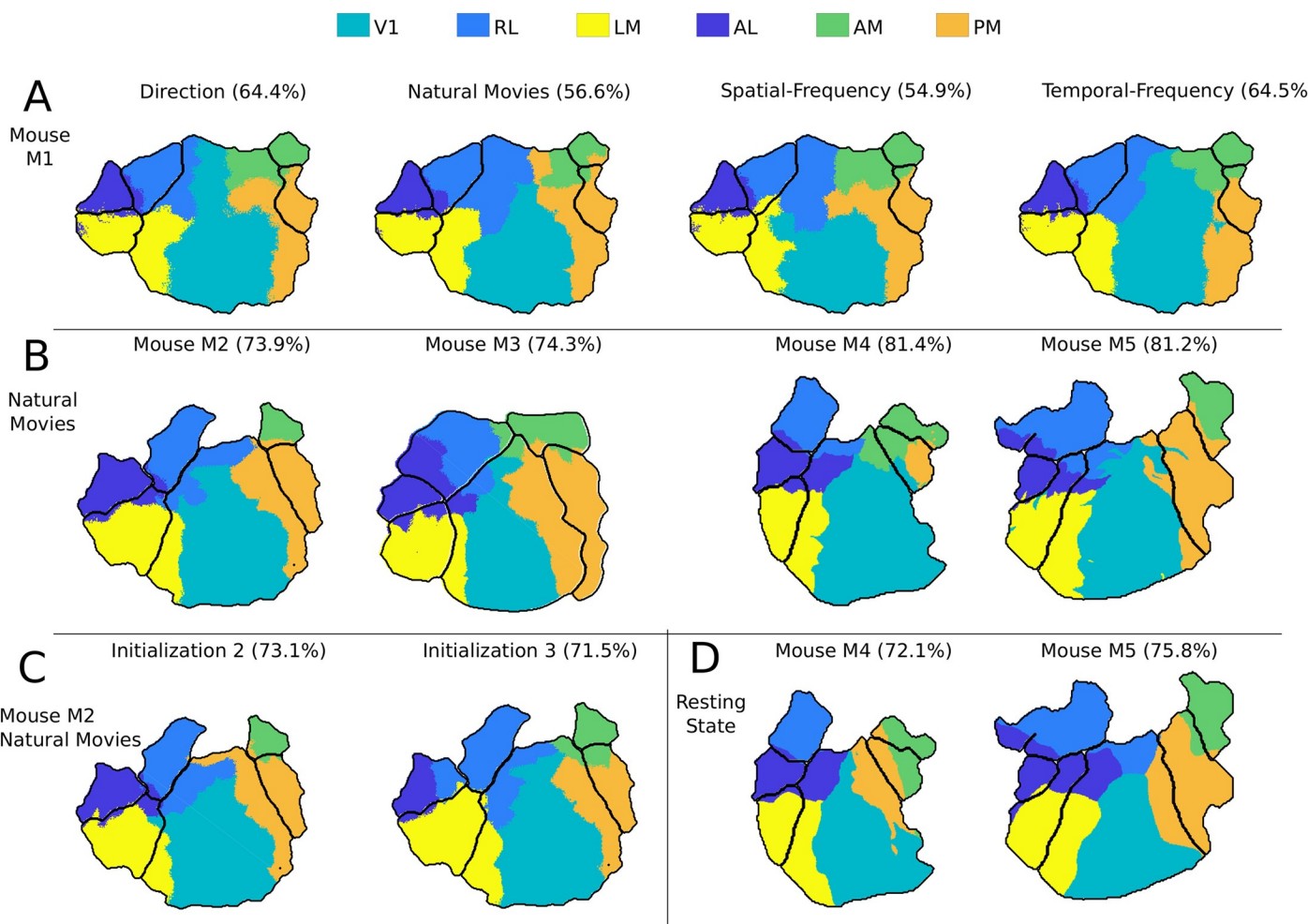

**Fig 6. Boundaries obtained by clustering visual cortex areas using the semi-supervised pipeline. A**) Boundaries derived using different visual stimuli in one mouse. Visual stimuli and accuracy (in %) are noted. **B**) Boundaries obtained for different mice using natural movies as stimuli. **C**) Boundaries obtained for different initializations of UBM for the same mouse and keeping visual stimuli unchanged. **D**) Boundaries derived from resting state responses.

The semi-supervised clustering resulted in a classification accuracy of about 70% (Table 5). Although the results are inferior compared to the supervised classification, the boundaries were observed to be close to the ground truth retinotopic boundaries in Fig 6. Irrespective of mice and visual stimuli, V1 was identified as the most significant visual area even though the same amount of labeled data was used from each area (Fig 6).

In Fig 6C, two different boundaries obtained by different random initializations of the UBM are shown. Although the accuracies obtained were different, the boundaries were consistent with each other. The clusters formed using the resting state responses were also consistent with the retinotopic maps (Fig 6D). This result provides additional support for our conjecture that there are intrinsic response characteristics of each visual area that generalize across stimuli and enable classification.

### 2.3 Resting state vs. stimulus-induced responses

In Sections 2.1 and 2.2, the natural movies and other stimuli were presented multiple times and averaged to obtain the stimulus-induced response. Since a trial structure cannot be defined for resting state responses, the dF/F of the signal was used directly as the input. In this section, we compare resting state responses with single-trial and trial-averaged stimulus-induced responses of various durations.

In Fig 7, the accuracies obtained by supervised and semi-supervised approaches are compared between the resting state responses, the single-trial responses of natural movies, and the trial averaged version of the same stimuli by varying the duration. All three results for the wide-field dataset are shown for responses varying from 4.5 secs to 54 secs (12 movies). In the two-photon dataset, the responses were sampled varying from 10 secs to 110 secs for natural movie 3 (120 secs) and spontaneous responses from "Session A" of the dataset.

Fig 7 shows that for both datasets, the results of stimulus-driven responses that were averaged across multiple trials had higher accuracy and asymptoted faster than the resting state responses. In the constrained semi-supervised clustering, the responses to natural movies were observed to classify the areas much better than the resting state responses. It is interesting to note that the supervised and semi-supervised approach was able to predict the boundaries accurately with just 4.5 secs of data. For the resting state responses, when the duration was increased to 800 secs (as in Sections 2.1 and 2.2), they also reliably clustered the visual areas, with accuracy close to natural movie responses. The same results were observed with the two-photon dataset as well. This result suggests that stimulus-driven responses contain better-discriminating responses compared to responses without an overt stimulus. Stimulus driven responses are trial averaged, thus have low intra-class variability. However, with resting state responses, a longer duration is required to model the intra- and inter-class variabilities.

Fig 7 also shows the result of using single-trial natural movie responses. In the wide field dataset, we observed that the single-trial responses give better accuracy than the resting state responses. However, with the two-photon dataset, the performance of spontaneous and single-trial movie responses was similar.

### 3 Discussion

In this paper, the responses of six visual areas of mouse cortex to various stimuli were studied using data-driven methods. First, in Section 2.1, the retinotopically defined boundaries were used as ground truth to train supervised classifiers, and the functional uniqueness of the visual areas was evaluated in terms of classification accuracy on held-out test data. The result indicates that the supervised models were able to discriminate retinotopic borders of different areas using any of the stimuli. Although limiting the training data to the center resulted in a

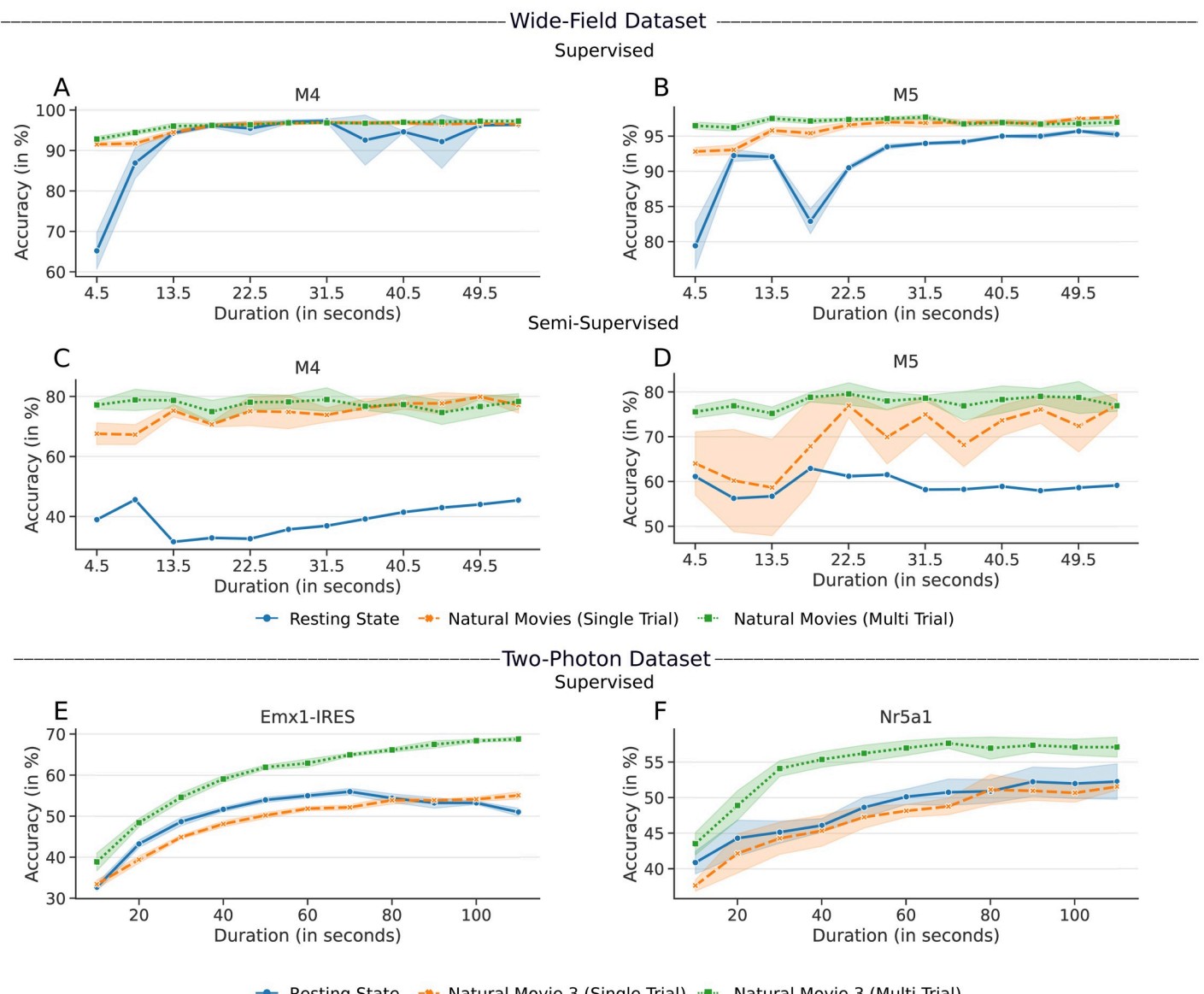

**Fig 7. Accuracies obtained by the supervised/semi-supervised pipeline with varying response lengths of resting state and stimulus-induced responses. A**, **B**) Results for Mouse M4 and M5, respectively, using the supervised approach. **C**, **D**) Results using the semi-supervised approach. **E**, **F**) Results for the two-photon dataset, using the Cre-lines Emx1-IRES and Nr5a1, respectively.

decrease in classification accuracy, the degradation in performance was graceful. It is to be noted that even the retinotopically defined boundaries can vary in detail between recording sessions [9]. Given these results of supervised classifiers trained on a range of stimuli, we conjectured that each area of the mouse brain has specific response patterns that reflect their underlying local and long-range input circuits. The scalability of the proposed approach on the dataset from the Allen Institute and the experiments on the resting state analysis (with no overt stimuli) adds to this hypothesis. To further test this hypothesis and to reduce the impact of the correlated dataset, the supervised information was minimized in Section 2.2 using a semi-supervised approach.

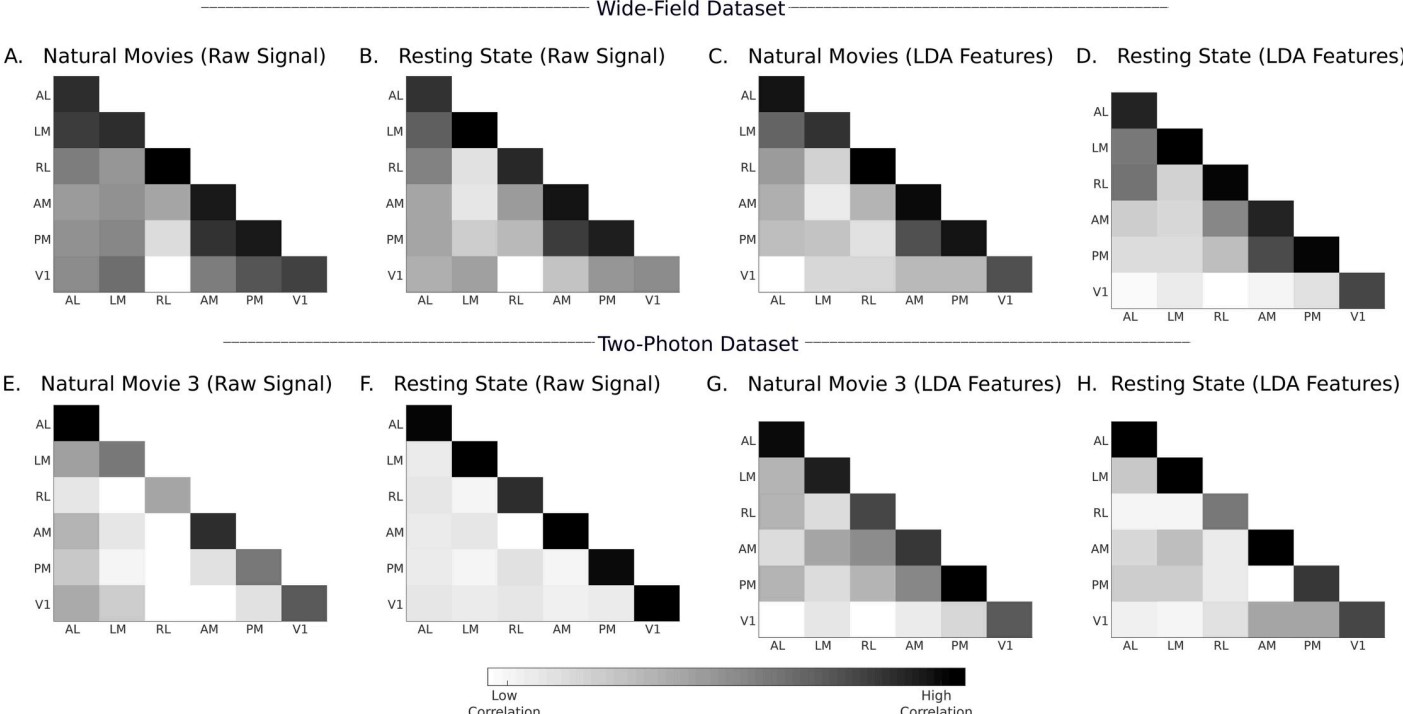

**Fig 8. Intra-area and inter-area correlations computed on input responses and LDA features. A–D**) Correlations computed from mouse M4 of wide-field dataset. **E–H**) Correlations computed from Emx1-IRES Cre-line of two-photon dataset. The correlations are computed as averages over all unique pairs of neurons/pixels in the test data, which were not used to train the LDA projection matrix. The correlations in the two-photon dataset are computed using data pooled from different mice and multiple sessions. In S3 and S4 Figs, we present a detailed correlation analysis with information about individual mice and session. Further examples of correlations analysis are shown in S5–S7 Figs.

In the semi-supervised approach, the center-most pixels were the only supervised information given to the algorithm. From the center-most cluster, each visual area was expanded iteratively. The accuracy of the final clustering was used as a measure of functional uniqueness (and activation pattern) of the visual areas. Even with this small amount of supervised information, the algorithm was able to find boundaries that were consistent with that of retinotopically obtained boundaries. This result, together with the resting state data, adds to the conclusion that visual areas have characteristic responses that can be used to classify their borders.

To analyze the nature of such response features, in Fig 8, we show the inter- and intra-areal correlations between the neuronal responses that were input into the pipeline. Fig 8A and 8B shows the correlation computed for wide-field data using natural movies and resting state responses, respectively. Similarly, Fig 8E and 8F shows the same result for data from the Emx1-IRES subset of the two-photon dataset. For both datasets, the average intra-area correlation is consistently higher than the inter-area correlation. Even with resting state responses, in the absence of overt visual stimuli, the responses are more correlated within an area. The ratio of average intra- to inter-area correlations calculated on these responses were 1.1 and 2.5 for the wide-field and two-photon datasets, respectively. These raw signals were preprocessed using PCA and LDA before they were given to the classifiers. In Fig 8C, 8D, 8G, and 8H, the correlations computed in the LDA domain are shown. The LDA space significantly improved the ratio of average intra-area and inter-area correlations to 8.8 and 9.7 for the wide-field and two-photon datasets, respectively. In Fig 9, we show the 2D visualization of LDA subspace

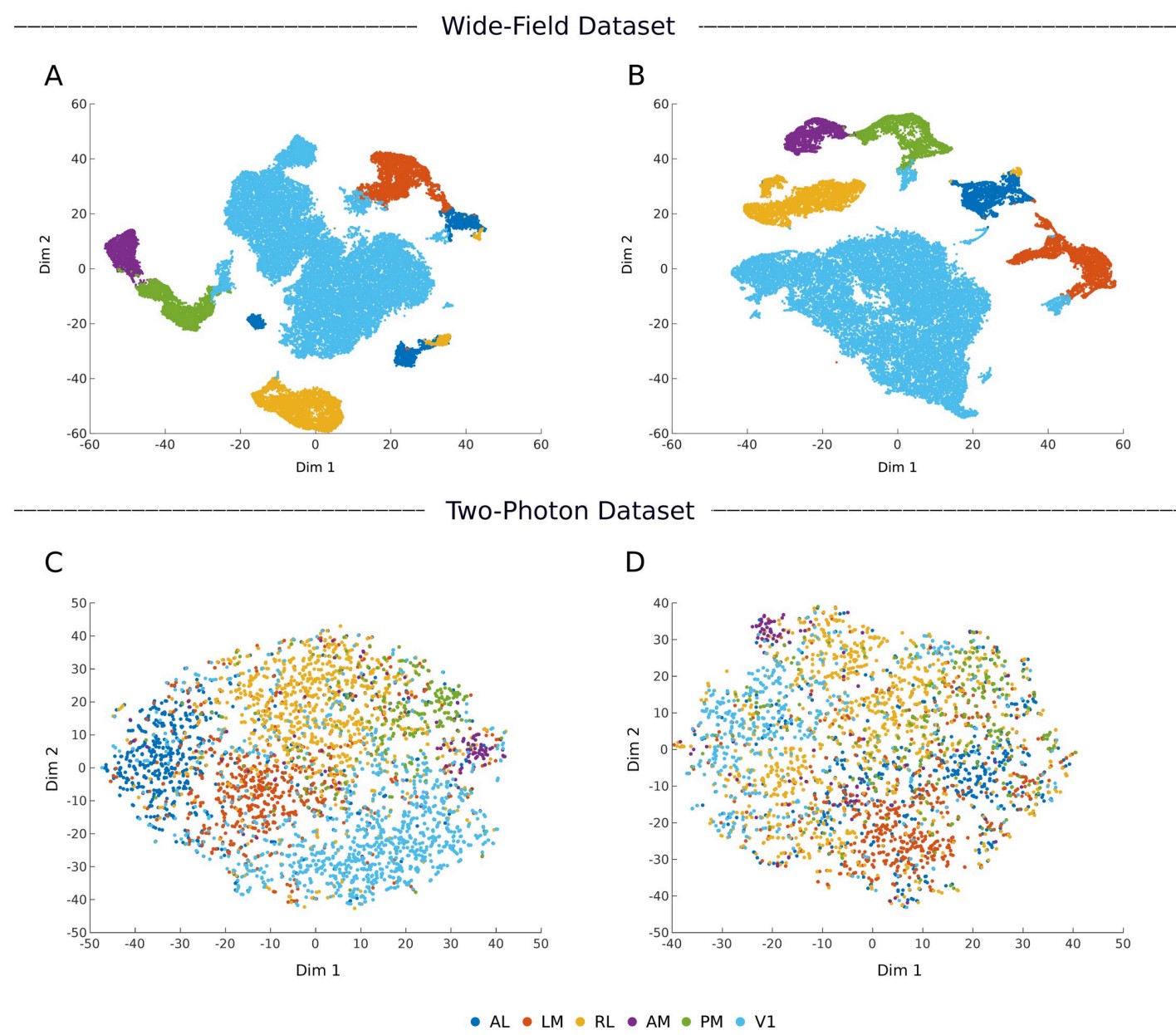

**Fig 9. Two-dimensional representation of the supervised LDA subspace. A, B**) LDA subspace of wide-field dataset (mouse M4). **C, D**) LDA subspace of two-photon dataset (Cre-line Emx1-IRES). The plots on the left (**A, C**) are obtained from natural movie responses and that on the right (**B, D**) are obtained from resting state responses.

obtained using wide-field (mouse M4) and two-photon (Emx1-IRES Cre-line) datasets. T-distributed stochastic neighbor embedding (tSNE, [44]) was used to convert the multi-dimensional LDA subspace into a visualizable 2D space. The LDA subspace is able to cluster neurons from different areas owing to correlated examples from the training data, in both visually driven and resting state responses from wide-field and two-photon datasets (Fig 9). By applying different supervised classifiers to these subspaces, we were thus able to identify the area labels with high accuracy.

The results obtained in Tables 3 and 4 are not based on a single classifier. We have shown the proposed supervised methods to work with generative (GMM, Unimodal Bayes) and discriminative (SVM, ANN) classifiers. We have also shown the approach to work with linear (Unimodal Bayes, SVM) and non-linear (ANN, GMM) classifiers. This shows that the obtained results are mainly because of the proposed dimension reduction using PCA and LDA rather than the classifier (Fig 9). In addition to this, the results in Tables 3 and 4 and Fig 4, show that the classification accuracy and confusion matrix are poor for the two-photon dataset when compared to the wide-field dataset. We note that the pixels in the wide-field dataset represent the pooled response of neurons, whereas the two-photon dataset captures individual neuron responses. There is higher variability in the neuronal responses, which leads to poorer performance. In addition, for the two-photon dataset, we selected all the neurons available for the given Cre-line enabling us to study all six areas. However, this hardly represents the entire population response from these areas, as only a few neurons were recorded from each area, and they were pooled from different mice (but see S3 and S4 Figs).

Our findings show that resting state responses are an important complement to visual responses in classifying areas. Visual as well as resting state responses do not arise de novo; rather they both reflect specific underlying input-output connections and circuits in each area. These connections can be activated by internally generated activity without overt visual stimuli, or explicitly by visual stimuli. We show comparable results from two different datasets, each with multiple visual stimuli along with resting state responses, demonstrating that resting state and visual responses can be used to classify area borders (Section 2.1). In Section 2.3, we further analyzed the difference between the resting state and stimulus-induced responses. The result shows that the responses averaged across multiple trials give better classification results than the single-trial or resting state responses for a fixed duration.

These findings demonstrate two important features of visual areas in mice, relevant to processing of visual stimuli. First, they are consistent with the fact that each cortical area is characterized by a unique pattern of connections and circuits. Some of these may be common to many areas of cortex (eg., local recurrent excitatory connections) whereas others are crafted by a combination of specificity and plasticity (eg., local inhibitory connections, long-range excitatory connections). These connection patterns come into play even with internally generated activity, which characterizes resting state responses. Thus, the intra-areal correlations are higher than inter-areal correlations for both visual and resting state responses (Fig 8). Second, visual stimuli are stronger drivers of internal circuits than resting state activity. Thus, the visually driven responses have higher classifier accuracy than resting state responses for given response durations, particularly when averaged across multiple trials, and in some instances the resting state responses never reach the accuracy of visual responses (Fig 7).

Motivated by the results obtained by supervised and semi-supervised classifiers, we attempted to cluster the pixels from the wide-field dataset without using any labeled data. It is a significantly under-constrained problem to arrive at the boundaries without any explicit retinotopic information. The clustering results were not well matched to retinotopically defined areas.

## 4 Conclusion

In this work, different machine learning techniques were explored to obtain the visual area boundaries of six cortical areas. The boundaries obtained by supervised models are highly consistent with the ground truth obtained by retinotopic imaging. The results of data-driven models degrade as the supervised information is removed. However, our critical observation is that data-driven models can classify these areas accurately based on responses to a range of visual stimuli, which extends as well to responses in the resting state. This result is consistent with

the presence of unique area-specific circuitry in the visual cortex, which shapes visually driven or resting state activity in these areas.

## Supporting information

**S1 Fig. Horizontal and vertical retinotopy along with sign map within six visual areas of all the mice used in the paper.** Cortical areas of the left hemisphere are shown. Azimuth 0˚ and 90˚ correspond to the midline and the far periphery of the contralateral visual field, respectively. Negative values of elevation represent lower visual field and positive values represent upper visual field.
(EPS)

**S2 Fig. Confusion matrices obtained using supervised GMM classifiers for all animals and data.** In Fig 4, confusion matrices were shown for an example mouse and a Cre-line from the wide-field and two-photon datasets, respectively. Here we show the confusion matrices for all mice and Cre-lines from the wide-field and two-photon datasets, respectively.
(EPS)

**S3 Fig. Intra-area and inter-area correlations computed from natural movies responses of individual mice from the two-photon dataset.** In Fig 8, intra-area and inter-area correlations were computed at the Cre-line level with data pooled from different mice and sessions for the two-photon dataset. Here we present the inter and intra-area correlations for various individual mice from the dataset, which had responses recorded from three or more sessions.
(EPS)

**S4 Fig. Intra-area and inter-area correlations computed from resting state responses of individual mice from the two-photon dataset.** In Fig 8, intra-area and inter-area correlations were computed at the Cre-line level with data pooled from different mice and sessions for the two-photon dataset. Here we present the inter and intra-area correlations for various individual mice from the dataset, which had responses recorded from three or more sessions.
(EPS)

**S5 Fig. Intra-area and inter-area correlation computed on input responses for all animals and data.** In Fig 8, average intra-area and inter-area correlations computed from input responses were shown for an example mouse and a Cre-line from the wide-field and two-photon datasets, respectively. Here, we show the correlations for all mice and Cre-lines from the wide-field and two-photon datasets, respectively.
(EPS)

**S6 Fig. Intra-area and inter-area correlation computed on LDA features for all animals and data.** In Fig 8, average intra-area and inter-area correlations computed from LDA features were shown for an example mouse and a Cre-line from the wide-field and two photon-datasets, respectively. Here, we show the correlations for all mice and Cre-lines from the wide-field and two-photon datasets, respectively.
(EPS)

**S7 Fig. Comparison of intra and Inter-area correlation computed using all pixels and patches of pixels with the same retinotopic map. A, C)** Intra and inter-area correlations computed using all pixels. **B, D)** Corresponding correlations computed using patches of pixels with approximately the same retinotopy from each area and averaged across the entire visual space. These data show that intra-area pixels are more correlated than inter-area pixels.
(EPS)

**S1 Table. Number of neurons available for other Cre-lines in the Allen Institute dataset.** In the text, we analyzed Emx1-IRES and Nr5a1 Cre-lines from dataset 2 (Section 1.2.2). Here we present the number of neurons available from all the other Cre-lines that contain all the six visual areas considered in the paper.
(PDF)

**S2 Table. Classification accuracy for other Cre-lines in the Allen Institute dataset.** In the text, we presented the classification accuracy for Emx1-IRES and Nr5a1 Cre-lines from dataset 2 (Table 4). Here we present the same for all the other Cre-lines in dataset 2 that contain all the six visual areas considered in the paper.
(PDF)

## Acknowledgments

We thank the Centre for Computational Brain Research (CCBR), IIT Madras for enabling the collaboration between Sur Lab, Massachusetts Institute of Technology and Indian Institute of Technology Madras.

## Author Contributions

**Conceptualization:** Mriganka Sur, Hema A. Murthy.

**Data curation:** Mari Ganesh Kumar, Ming Hu.

**Formal analysis:** Mari Ganesh Kumar, Ming Hu, Aadhirai Ramanujan.

**Funding acquisition:** Mriganka Sur, Hema A. Murthy.

**Investigation:** Mriganka Sur, Hema A. Murthy.

**Methodology:** Mari Ganesh Kumar, Ming Hu, Aadhirai Ramanujan, Mriganka Sur, Hema A. Murthy.

**Project administration:** Mriganka Sur, Hema A. Murthy.

**Resources:** Mriganka Sur, Hema A. Murthy.

**Software:** Mari Ganesh Kumar, Ming Hu.

**Supervision:** Mriganka Sur, Hema A. Murthy.

**Validation:** Mriganka Sur, Hema A. Murthy.

**Visualization:** Mari Ganesh Kumar, Ming Hu.

**Writing – original draft:** Mari Ganesh Kumar, Ming Hu.

**Writing – review & editing:** Mari Ganesh Kumar, Ming Hu, Mriganka Sur, Hema A. Murthy.

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
