## [Decision Letter · Decision Letter 0]

5 Jun 2020

Dear Mr. Kumar,

Thank you very much for submitting your manuscript "Functional Parcellation of Mouse Visual Cortex Using Statistical Techniques Reveals Clustering and Signatures of Cortical Processing Areas" for consideration at PLOS Computational Biology.

As with all papers reviewed by the journal, your manuscript was reviewed by members of the editorial board and by several independent reviewers. In light of the reviews (below this email), we would like to invite the resubmission of a significantly-revised version that takes into account all of the reviewers' comments.

It is particularly important that you attend to the questions the reviewers raised about interpretations/utility. Reviewer 1 argues that, as it stands, the paper provides no insight into visual processing, since similar clusters are derived from resting state activity and no information is given on the signatures of visual responses of different areas. Reviewer 3 made a similar comment, noting that you suggest that each visual area has distinct response signatures, but don't really provide much insight into the nature of those signatures, particularly given that the classifiers can separate these visual areas using just 10 minutes of resting state activity. You should be very sure to address these fundamental concerns regarding interpretation and utility of the study.

We cannot make any decision about publication until we have seen the revised manuscript and your response to the reviewers' comments. Your revised manuscript is also likely to be sent to reviewers for further evaluation.

Sincerely,

Blake A. Richards

Associate Editor

PLOS Computational Biology

Lyle Graham

Deputy Editor

PLOS Computational Biology

It is particularly important that you attend to the questions the reviewers raised about interpretations. Reviewer 1 argues that, as it stands, the paper provides no insight into visual processing, since similar clusters are derived from resting state activity and no information is given on the signatures of visual responses of different areas. Reviewer 3 made a similar comment, noting that you suggest that each visual area has distinct response signatures, but don't really provide much insight into the nature of those signatures, particularly given that the classifiers can separate these visual areas using just 10 minutes of resting state activity. You should be very sure to address these fundamental concerns regarding interpretation and utility of the study.

Reviewer's Responses to Questions

**Comments to the Authors:**

Reviewer #1: Kumar et al. studied wide-field GCaMP signals in 6 cortical visual areas in the mouse. Pixels clustered into 6 groups with boundaries that match retinotopic borders, indicating that each visual area is distinct. Unfortunately, what causes these areas to cluster is unknown. With no information on the basis of clustering it’s difficult to assign significance to the clusters. Furthermore, the distinguishing feature is not related to visual stimuli since the same clusters were derived from spontaneous activity, suggesting that the clusters contain no information about the roles of these areas in processing visual information. In short, the paper reports an observation – that pixels cluster – but doesn’t explain why. The observation would appear to have little significance.

Specific comments

(1) The paper’s littered with sentences that appear disconnected from the results and conclusions. I’ll give two examples. The second sentence of the abstract reads: ‘The extent to which these areas represent discrete processing regions remains unclear.’ The paper brings no clarity. Why suggest the study is of visual processing? And in the results ‘These results indicate that each visual area has a characteristic signature that is represented in the responses to visual stimuli presented and can be revealed with a variety of visual stimuli.’ What signature? Information on this signature is conspicuously missing. And about the only thing we know is that the signature’s not related to presented visual stimuli. This sentence is comprehensively at odds with the results.

The text frequently suggests the paper will be about the different visual stimuli that drive these visual areas, but there’s no information here on this topic. My sense is that the study didn’t lead in the direction the authors had expected. If so, it would be best to let go of the intended direction of the project and address what they can with the results. In particular, the aim and conclusions need to be clear and clearly related to the results.

(2) 1.1.1 What is the genotype of the transgenic GCaMP mice? Also, the breeding scheme.

(3) What’s the eye-to-screen distance and visual angle subtended by the monitor.

(4) What was the size (in degrees) of the visual stimuli?

(5) Figure 1D. Why does the field sign map appear so patchy? It’s different from the maps produced by others with wide-field GCaMP imaging. See for example Zhuang et al. eLife 2017. I would guess perhaps the SNR of the retinotopic maps are poor?

(6) ‘The boundaries of the 6 core visual areas were defined according to criteria described in [9]’ This statement is conspicuously untrue. If the authors intended to replicate the field sign mapping technique of Garrett et al., they have failed. In Garrett et al. the borders are, by definition, where the field sign crosses zero. In figure 1D, the borders are not at zero. Many, perhaps all the borders are at some negative field sign value. The authors need to take a closer look at their code. They also need to provide a more detailed explanation of their mapping procedure and, ideally, the code.

(7) Table 1 provides incomplete information on the stimuli. For 1, give the SF and TF. For 2, the TF and direction. For 3, the SF and direction. We also need luminance, contrast and size of the stimulus.

(8) 2.1 The supervised classifiers are described in numerical detail, but I gained no insight into the differences between the classifiers. Why these classifiers? Was there reason to use several?

(9) Figure 4D and E. The LM and RL labels are swapped.

(10) Figure 5 and later figures. Why are the maps broken down by visual stimulus when earlier results were not. And why break them down by visual stimulus when clustering needs no visual stimulus?

(11) 2.3 Resting vs stimulus induced response. I failed to grasp the aim of this section.

(12) The unsupervised clustering simply fails. Is there a reason to include it? The stated conclusions are weak at best.

Reviewer #2: This paper addresses the question of whether the retinotopic visual cortical areas of the mouse can be discovered from their activity patterns in response to visual stimuli or in the resting state. The study concludes that retinotopically defined areas have unique activity profiles that allows their identification based on supervised and semi-supervised methods. However, unsupervised approaches fail to recover these areas with great accuracy, suggesting that despite differences between areas, there is also a great deal of overlap between areas. The question posed is an interesting one, and overall the results are convincing. I like the general approach and think this approach will be valuable for many future questions, even beyond studies of visual cortex. I therefore support publication of this work if the points below can be addressed.

Specific comments

1. In the discussion and conclusions, a lot of emphasis is placed on the resting state data. The authors emphasize that some of the separation between areas could be due to intrinsic activity rather than visual responses. However, there are only two mice for the resting state data, which seems like too small of a sample size. Either these claims should be lessened or more resting state data should be added.

2. I like the analysis of shuffling the area labels in the supervised analysis to show the chance level. I think similar analyses would be nice for the other parts of the paper too. For example, for the unsupervised clustering, it might be interesting to compare clustering metrics for the real data and data in which the pixel locations are shuffled. This could provide some measure of how much structure can be discovered in the real data relative to what would emerge from random data. In general these comparisons are helpful to provide the reader with a bound on what can be expected by chance.

3. My understanding is that the unsupervised analysis was only performed on the widefield calcium imaging data. It was a bit hard to figure this out in the text, so I apologize if I am incorrect. If my statement is correct, then it would be nice in addition to see the unsupervised analysis on the single cell data. The single cell data lack spatial correlations that are present in the widefield data, as the authors note. It would be interesting to see if similar clusters could be uncovered with the single cell data.

4. The tables of accuracies for the supervised and semi-supervised analyses are nice, but it would also be interesting to see the confusion matrices for these analyses. It would be interesting to some readers to see which areas are more similar to one another and thus get confused with one another more frequently. Such a confusion matrix could support some of the claims about lateral versus medial differences that the authors make using the unsupervised analysis.

5. The raw retinotopic map data for all mice should be shown in addition to the post-processed boundaries. This is important to evaluate the quality of the input to parcellate the areas into retinotopic divisions. In particular I ask about this because I was surprised by how much variance there was in the size and location of the areas. For example, in some mice AM is anterior and medial to PM, whereas in other mice AM is anterior and lateral to PM. Also, sometimes AM is directly bordering V1, and other times it is not. I was surprised by the location of AM in Figure 1D. Typically AM is adjacent to V1. Similar variance is seen for other areas. I am not sure it matters greatly for this study, but I have some concern that the area labels may be inaccurate in some mice, such as the case for AM that is not adjacent to V1 (Figure 1D).

6. Currently all the analyses are done within a mouse, which is sensible. However, I was wondering if the authors tried across mouse analyses. For the supervised analyses, what do the results look like if the classifiers are trained on mouse 1 and tested on mouse 2? For the unsupervised analysis, is there a way to see if the clusters identified in mouse 1 then provide predictive power for mouse 2? Across mouse analysis might further support claims of structure made by the authors.

7. What were the mice doing during the imaging experiments? Were they moving? Could movement contribute to the results? Recent studies have emphasized the importance of movement to visual cortical activity (see PMIDs: 20188652, 31551604, 31000656).

8. Some references to recent papers using related methods were missing and should be added. PMIDs: 32282806, 30772081.

Reviewer #3: The authors use widefield and 2-photon imaging data from the mouse visual cortex to train classifiers to identify the different visual areas. They find that supervised and semi-supervised classifiers perform well, identifying pixels or neurons with high accuracy. The fact that they do so using even just the neural responses to one 4.5 second movie is remarkable. The authors go on to show that unsupervised classifiers do not identify the different visual areas with high accuracy, but do capture some of the functional organization of the visual cortex. These results indicate that there are distinct physiological profiles for the different visual areas – or from the unsupervised results at least from groups of visual areas. I found the work to be interesting, and the paper did a great job of explaining the different techniques and conclusions. I do have some concerns, that I hope are reasonably addressable.

Major Concerns:

1. The retinotopic maps shown in these figures are somewhat different from the retinotopic maps I see in the literature (namely Zhuang et al 2017 and Garrett et al 2014) – specifically in regards to the location and borders of RL. In the two papers mentioned above, RL sits at the top of V1, in contact with both AL and AM on either side. I understand that this area has difficult retinotopy, however, it seems possible that the pixels at the top of V1 are being mis-assigned to V1 and should really be within RL – which could had different effects on the different supervised/semi-supervised/un-supervised results. I encourage the authors to look more closely at the assignment for RL, or perhaps to consider excluding it from these analyses (or weighting the accuracy for that area differently).

2. I am wary of the conclusions drawn from the unsupervised classification results. Specifically, it seems that the conclusions drawn result directly from the rules added to the unsupervised clustering. For instance, since clusters can only be merged if they are touching, it seems impossible for the lateral and medial areas to end up in the same clusters, especially given the 40% constraint, so it isn’t clear to me how meaningful that result is. It is possible that the paper could stand without the unsupervised classification results.

3. Throughout the paper, chance is said to be 1/6 given the six visual areas. It is not clear to me that this is the right level of chance to be used. Particularly for the widefield data, when more pixels are in V1 than any of the other visual areas, it seems that the prior should be shifted towards V1. Is there a way to define chance that takes the relative proportion of pixels (and neurons for the 2P data) for each of the areas?

4. Related to this, I would like to see what the confusion matrix looks like for these classifications. Do mis-classified pixels(/neurons) tend to be classified as the closest area? To the area the best matches retinotopy for that location? Or do they default to V1?

5. I would like to see a comparison of the semi-supervised area boundaries with retinotopy (eg. Fig 9 of Zhuang et al). It is not clear to me that the boundaries that this method is identifying do not reflect retinotopy. Eg. It appears that the semi-supervised boundaries separate altitude reasonably well (eg. the boundary between LM and RL that extends into V1 seems to match roughly with the horizontal meridian). The authors make the point that they are not using a retinotopic stimulus, but a natural movie stimulus has distinct information in different retinotopic locations – and thus could drive retinotopically distinct responses. If that is not true for the movies used in this study, I’d like to see an analysis to demonstrate the spatial/temporal content of the movies across retinotopy.

6. The biggest question that emerges for me from this work is what is the distinguishing features of the activity from the different areas. The authors conclude that these results suggest that each visual area has distinct response signatures, and some insight into the nature of those signatures would be very valuable. Particularly given that these classifiers can separate these visual areas using as little as a 4.5 second movie or even just 10 minutes of resting state activity. What are the features of the activity that the classifiers are using to separate these areas? An analysis of the classifier weights or features could be really illuminating in this regard. Or perhaps even example traces from the different areas.

7. The analysis in Figure 6 comparing boundaries obtained with different durations of stimulus is very interesting and important. My concern is that the movie responses are averaged across trials while the resting state is not averaged. The nature of an averaged signal and an unaveraged signal is very different, so 20 seconds of average movie activity and 20 seconds of unaveraged resting state are not an equivalent comparison. Why not do the classification of movie responses without averaging the trials, thus allowing a direct duration comparison between the two?

Minor concerns:

1. There are a lot of details regarding the methods that are missing from the manuscript. Each of these is minor, but altogether I do consider this important:

• Table 1 summarizing the stimuli used for widefield imaging needs more information. Namely the spatial and temporal frequencies and contrast used for stimulus 1. The directions, temporal frequency, and contrast for stimulus 2. The directions, spatial frequency, and contrast for stimulus 3.

• Was the stimulus for the widefield imaging warped to account for viewing distance? Where was the monitor positioned relative the mouse’s center of gaze? As different visual areas cover different regions of retinotopy, if the stimulus wasn’t warped properly, the stimulus could have different content in different regions, and hence for different HVAs.

• What was the mean luminance of the stimulus? Was the monitor gamma corrected?

• What Cre line was used to drive the GCaMP6 expression in the widefield data?

• How did the authors choose to analyze Emx1 and Nr5a1 from the Allen Brain Observatory dataset? (the authors mention these Cre lines were imaged across all six areas, but that is true for Cux2, Rorb, and Rbp4 as well – why were these not analyzed?) Was Emx1 used from all layers or only from specific layers?

• I’d like a bit more information about how the classifier was applied to the 2P data. Were all neural traces (for the chosen Cre lines) used, or subselected? If subselected, how was this done? What was the test/train split? Were equal numbers of neurons used for each area or was this different? How many neurons were used?

2. Figure 3B shows generalization across mice. It’s not clear to me whether this is to show similar results for different mice, or whether it is to show that training on one mouse can predict testing on a different mouse. I believe it’s the former, but it would be very interesting if it were the latter. Please clarify.

3. Figure 4 color labels appear to be mis-assigned

4. Why is the accuracy for widefield pixels so much higher than for the 2P neurons? Given the shorter movie clip, and the single pixel data, I’d expect the widefield data to perform worse than 2P, not better. But perhaps the fact that the widefield signal for a pixel could combine activity from multiple neurons/processes could play a role in this? In a similar vein, why do Emx1 and Nr5a1 perform differently? Are there different numbers of neurons available? Could it be layer specific? I don’t think these can necessarily be conclusively answered, but if possible some discussion of these questions would help.

5. The Allen Brain Observatory is from the Allen Institute for Brain Science (not Allen Brain Institute – line 42). The citation should also match the citation policy for the dataset (https://alleninstitute.org/legal/citation-policy/)

**Have all data underlying the figures and results presented in the manuscript been provided?**

Reviewer #1: Yes

Reviewer #2: Yes

Reviewer #3: Yes

PLOS authors have the option to publish the peer review history of their article (what does this mean?). If published, this will include your full peer review and any attached files.

Reviewer #1: No

Reviewer #2: No

Reviewer #3: No
---

## [Decision Letter · Decision Letter 1]

8 Oct 2020

Dear Mr. Kumar,

Thank you very much for submitting your manuscript "Functional Parcellation of Mouse Visual Cortex Using Statistical Techniques Reveals Response-Dependent Clustering of Cortical Processing Areas" for consideration at PLOS Computational Biology. As with all papers reviewed by the journal, your manuscript was reviewed by members of the editorial board and by several independent reviewers. The reviewers appreciated the attention to an important topic. Based on the reviews, we are likely to accept this manuscript for publication, providing that you modify the manuscript according to the review recommendations. Specifically, Reviewer 1 has some remaining concerns about your border analyses and Reviewer 3 has several remaining significant concerns regarding your interpretations of your analyses. If you can address these concerns then we will likely be able to accept this paper.

Sincerely,

Blake A. Richards

Associate Editor

PLOS Computational Biology

Lyle Graham

Deputy Editor

PLOS Computational Biology

[LINK]

Reviewer's Responses to Questions

**Comments to the Authors:**

Reviewer #1: Uploaded as attachment.

Reviewer #2: The authors have adequately addressed my comments.

Reviewer #3: I think the authors have made some significant improvements to this paper that strengthen and clarify the work. There are, however, some outstanding issues I think need to be addressed

The addition of the confusion matrix in Figure 4 is valuable and informative – thank you for adding that. It might be worth pointing out that for the widefield data, ignoring V1, there is a higher (though not high) confusion of AL and LM for each other, and of AM and PM for each other. To me, this is interesting, and is consistent with other studies that show these two pairs of areas have more similar responses with each other that with other areas. This might be worth mentioning as I believe that it supports the idea that the clustering here represents functional differences.

The fact that this is not clear in the 2P data, however, might make this harder to support. Could you comment on why the confusion is greater for the 2P than the widefield? Is it that there is diversity of functional responses of neurons, so when looking at individual neurons the confusion will be higher than pixels that are merging signals from multiple cells?

Another question I have from the confusion matrix is that in the 2P data particularly, areas AM and PM appear to have lower accuracy than the other areas. Perhaps this is more true for the Emx1 data than the Nr5a1 data (the lower and more variable # of neurons in the Nr5a1 data makes me a bit wary of over-interpreting those results). Could you speak to why this might be?

I continue to disagree with the authors’ claim that the natural movie stimulus does not contain retinotopic information. This is not true. For any given frame of the movie, there is different content in different retinotopic regions. This might average out across all the frames so that the spatial/temporal frequency content is similar across space, but for each frame it is different. This is also true of the stimulus used for retinotopic mapping. While across the entire stimulus the content across space is the same, for each frame it is different – and THIS is what enables the authors to use the stimulus to map the retinotopy of the mouse visual cortex. But the same feature in the movies, different content in different regions of space for each individual frame, means that the movies do contain retinotopic information. The authors claim the opposite which is not true. This must be fixed.

And I still would like to see the comparison of the semi-supervised clustering with retinotopy (eg. Fig 9 of Zhuang et al) as the area borders appear to match retinotopy better than the area boundaries (that are derived from retinotopy).

I think Figure 7 improves the previous analysis of stimulus duration and enables a better comparison between the movie and resting state results. I think the authors could use this to emphasize two things in their text:

First, one of the stunning things from this analysis is that only a very short movie is needed to get fairly accurate separation of visual areas in the widefield data. 4.5 seconds contains probably only a few hundred stimulus frames, and they are likely fairly correlated frames at that. The fact that just a few visual stimulus features can drive this level of accuracy is, to me, really surprising, and I think this could be emphasized more in the text.

Second, the difference in the results here for supervised vs semi-supervised, and between widefield and 2p datsets, could be valuable for helping to understand the differences between these datasets. Eg. The fact that 2p accuracy is higher for averaged movie responses than single trial, while widefield shows little difference between them, could point to the impact of the correlated pixels on the results (eg. averaging across pixels vs averaging across trials). I think the fact that the supervised clustering (both widefield and 2p) has similar performance for resting state and single trial movie, while semi-supervised shows a difference between the two, might be similarly revealing for understanding the differences in those methods/results.

I find the new analysis in section 3 (figures 8 &9) very confusing and I don’t know that it helps to address the question that I had in my previous review.

First, the comparison of inter- and intra-areal correlations for the 2P dataset is fraught with problems. Namely, for the inter-area correlations, it must be pointed out (based on my understanding of the 2P dataset) that the areas are imaged in different sessions and different mice, while the intra-areal correlations include at least some data collected within the same session. This will make the intra-areal correlations higher simply because factors such as running or brain state will be in common. And while the correlations in the context of the movie at least have a common stimulus, the correlations of resting state … I just don’t know how to think about that comparison of inter- and intra-areal correlations. I would recommend the authors not include the 2P dataset in this particular analysis because of these issues.

However, my larger concern is that this analysis does not address my original question of what features of the activity are that separate these areas. The authors claim in their reply that this analysis shows that the intra-correlations are the key features. First, this analysis doesn’t convince me of this. I fully expect neighboring pixels/neurons to be more correlated, if only because they are retinotopically adjacent and thus likely receive common inputs, etc. Even in the absence of a patterned stimulus, they should have more correlations. I’d be more convinced if the inter- vs intra- areal correlations took retinotopy into account – eg. compared the same regions of retinotopy across areas.

However, even then, if the answer that the authors have for what features of activity separate the areas is just higher correlations within areas than across areas, it doesn’t tells us what’s different about, eg., AL and AM. The authors make the conclusion that “visual cortical areas have characteristic activity patterns” – and to me this statement is not supported by this comparison of correlations. It’s not clear to me that this type of analysis is unable to address the question of what distinguishes the areas, in which case I think the authors should walk back such claims.

Minor:

• Please indicate whether “resting state” for the wide field dataset is in the dark or with a gray screen?

• The wide-field dataset methods say the mice are Ai93 and that all mice expressed GCaMP6f or GCaMP6s. Ai93 is exclusively GCaMP6f, so either the “or GCaMP6s” is a mistake, or another reporter line (possibly Ai94) was used? Please fix.

• Mice expressing Emx1-IRES-Cre;Ai93 have been shown to have aberrant activity (Steinmetz et al 2017) – large cortex wide events. Do you think this could impact your results? In my mind, it seems unlikely for two reasons – one my recollection of the paper was that the aberrant activity was weaker in visual cortex than other areas; second, I would expect it to make pixels more similar to one another, so the fact that you are able to distinguish areas indicates that any aberrant activity likely isn’t factoring into the clustering analysis. It might be worth adding a comment on this – but I leave that to the author’s discretion.

• Are you using the ∆F/F traces for the 2P dataset provided through the AllenSDK or the raw fluorescence? I would assume the former, but there are mentions of “raw neuronal activity” that make me wonder. This would be important to add to the methods.

• The citation for the Allen Brain Observatory should not be Lein et al. 2007, but should be de Vries, Lecoq, Buice et al 2020.

**Have all data underlying the figures and results presented in the manuscript been provided?**

Reviewer #1: Yes

Reviewer #2: Yes

Reviewer #3: Yes

PLOS authors have the option to publish the peer review history of their article (what does this mean?). If published, this will include your full peer review and any attached files.

Reviewer #1: No

Reviewer #2: No

Reviewer #3: No
---

## [Editor Report · Decision Letter 2]

17 Nov 2020

Dear Mr. Kumar,

We are pleased to inform you that your manuscript 'Functional Parcellation of Mouse Visual Cortex Using Statistical Techniques Reveals Response-Dependent Clustering of Cortical Processing Areas' has been provisionally accepted for publication in PLOS Computational Biology.

Best regards,

Blake A. Richards

Associate Editor

PLOS Computational Biology

Lyle Graham

Deputy Editor

PLOS Computational Biology

---

## [Editor Report · Acceptance letter]

27 Jan 2021

PCOMPBIOL-D-20-00156R2 

Functional Parcellation of Mouse Visual Cortex Using Statistical Techniques Reveals Response-Dependent Clustering of Cortical Processing Areas

Dear Dr Kumar,

I am pleased to inform you that your manuscript has been formally accepted for publication in PLOS Computational Biology. Your manuscript is now with our production department and you will be notified of the publication date in due course.

With kind regards,

Alice Ellingham
